# Capsule type defines the capability of *Klebsiella pneumoniae* in evading Kupffer cell capture in the liver

Xueting Huang[1,2], Xiuyuan Li[1], Haoran An[1,2], Juanjuan Wang[1,2], Ming Ding[1], Lijun Wang[1,3], Lulu Li[1], Quanjiang Ji[4], Fen Qu[5,6], Hui Wang[7], Yingchun Xu[8], Xinxin Lu[9], Yuan He[10], Jing-Ren Zhang[1,2]*

1 Center for Infectious Disease Research, Department of Basic Medical Science, School of Medicine, Tsinghua University, Beijing, China, 2 Tsinghua-Peking Center for Life Sciences, Tsinghua University, Beijing, China, 3 Beijing Tsinghua Changgung Hospital, Tsinghua University, Beijing, China, 4 School of Physical Science and Technology, Shanghai Tech University, Shanghai, China, 5 The Center of Clinical Diagnosis Laboratory, 302 Hospital of PLA, Beijing, China, 6 China Aviation General Hospital of China Medical University, Beijing, China, 7 Department of Clinical Laboratory, Peking University People's Hospital, Beijing, China, 8 Department of Clinical Laboratory, Peking Union Medical College Hospital, Chinese Academy of Medical Sciences, Beijing, China, 9 Department of Clinical Laboratory, Beijing Tongren Hospital, Capital Medical University, Beijing, China, 10 Research Beyond Borders, Boehringer Ingelheim (China), Shanghai, China

☯ These authors contributed equally to this work.
* zhanglab@tsinghua.edu.cn

**Data Availability Statement:** The complete genomes of *K. pneumoniae* TH12908, TH12852, TH12880, TH12845 and TH12846 were uploaded

## Abstract

Polysaccharide capsule is the main virulence factor of *K. pneumoniae*, a major pathogen of bloodstream infections in humans. While more than 80 capsular serotypes have been identified in *K. pneumoniae*, only several serotypes are frequently identified in invasive infections. It is documented that the capsule enhances bacterial resistance to phagocytosis, antimicrobial peptides and complement deposition under *in vitro* conditions. However, the precise role of the capsule in the process of *K. pneumoniae* bloodstream infections remains to be elucidated. Here we show that the capsule promotes *K. pneumoniae* survival in the bloodstream by protecting bacteria from being captured by liver resident macrophage Kupffer cells (KCs). Our real-time *in vivo* imaging revealed that blood-borne acapsular *K. pneumoniae* mutant is rapidly captured and killed by KCs in the liver sinusoids of mice, whereas, to various extents, encapsulated strains bypass the anti-bacterial machinery in a serotype-dependent manner. Using capsule switched strains, we show that certain high-virulence (HV) capsular serotypes completely block KC's capture, whereas the low-virulence (LV) counterparts confer partial protection against KC's capture. Moreover, KC's capture of the LV *K. pneumoniae* could be *in vivo* neutralized by free capsular polysaccharides of homologous but not heterologous serotypes, indicating that KCs specifically recognize the LV capsules. Finally, immunization with inactivated *K. pneumoniae* enables KCs to capture the HV *K. pneumoniae*. Together, our findings have uncovered that KCs are the major target cells of *K. pneumoniae* capsule to promote bacterial survival and virulence, which can be reversed by vaccination.

to Genbank, under bioproject PRJNA778913 and
PRJNA846980.

**Funding:** This work was supported by grants from
National Natural Science Foundation of China
31820103001, 31530082, 81671972, 31728002
(J.-R.Z.) and 3210010245 (J.W.), Tsinghua
University Spring Breeze Fund 20201080767 (J.-R.
Z.), the China Postdoctoral Science Foundation
2020M680518 (J.W.), the Tsinghua-Peking Joint
Center for Life Sciences Postdoctoral Foundation
(X.H., J.W.), and grant from Boehringer Ingelheim
(J.-R.Z.). The funders had no role in study design,
data collection and analysis, decision to publish, or
preparation of the manuscript.

**Competing interests:** The authors have declared
that no competing interests exist.

## Author summary

*Klebsiella pneumoniae* is a major human pathogen. While capsule is the main virulence
factor of the pathogen, only several of more than 80 capsule serotypes are frequently iden-
tified in invasive infections. However, it remains unclear how capsule contributes to *K.
pneumoniae* virulence. Here we show that capsule type defines *K. pneumoniae* virulence
by differential escape of immune surveillance in the liver. While low-virulence (LV) types
are captured by Kupffer cells (KCs), high-virulence (HV) types circumvent the anti-bacte-
rial machinery. Further, inactivated *K. pneumoniae* vaccine enables KCs to capture the
HV *K. pneumoniae* and protects mice from lethal infection. Our findings explain the clini-
cal prevalence of HV capsule types, and provide promising insights for future vaccine
development.

## Introduction

*Klebsiella pneumoniae* is an encapsulated Gram-negative bacterium and a major pathogen of
pneumonia, urinary tract infection, liver abscess and sepsis [1]. The importance of *K. pneumo-
niae* in human health has been boosted by the emergence of genetic variants in the recent
decades, which are extremely resistant to many antimicrobials (classical Kp or cKp) or highly
virulent to healthy individuals (hypervirulent Kp or hvKp) [2]. cKp strains are typically iso-
lated from nosocomial infections of immunocompromised patients and frequently resist to
carbapenem, a major drug for treatment of Gram-negative pathogens. In a somewhat opposite
pattern, hvKp variants are associated with community-acquired invasive infections in healthy
individuals, and are mostly sensitive to carbapenem. *K. pneumoniae* virulence has been attrib-
uted to polysaccharide capsule, lipopolysaccharide (LPS), pili and siderophores, which exist in
virtually all virulent strains [1]. The hypervirulence traits of hvKp strains are associated with
overproduction of capsule [3, 4], additional siderophores [5], tellurite resistance [6, 7] and
hypermucoviscosity [8–10].

Capsule is an outmost layer and virulence factor of *K. pneumoniae* and many pathogenic
bacteria, which is considered as a protective structure against host innate immunity [11–13].
*K. pneumoniae* produces more than 80 types of polysaccharide capsule [14, 15]. Each type
structurally differs from the others in the repeating polysaccharide unit of capsular polysaccha-
ride (CPS) [12]. All known capsule variants of *K. pneumoniae* are synthesized by the Wzx/Wzy
polysaccharide polymerization machinery encoded by a single capsule polysaccharide (*cps*)
biosynthesis locus [14, 16]. Capsules are known for the ability to promote immune evasion
and survival of encapsulated bacteria [12]. The known mechanisms include hindering host
phagocytic clearance by repelling overall negative-charged phagocytes with similarly charged
CPS [17–19], masking the cell wall structures and membrane-associated proteins from com-
plement-mediated opsonophagocytosis [20–22], antigenically mimicking host glycans [23–25]
or enhancing bacterial resistance to oxidative killing [26]. However, it is virtually unknown
how the capsule promotes bacterial pathogenesis in the process of *K. pneumoniae* infection.

*K. pneumoniae* capsule types have shown great variations in virulence traits. hvKp isolates
are limited to capsule types of K1, K2, K16, K28, K57 and K63 [12, 27]. In particular, K1 and
K2 serotypes have been associated with approximately 70% of all hvKp infection cases reported
globally [28–31]. Accordingly, K1 and K2 strains of *K. pneumoniae* are more virulent than
many other capsule types in mice [32, 33]. However, it has not been defined if the differences
in capsular structure or type of *K. pneumoniae* are responsible for the serotype-dependent

clinical prevalence and virulence in mice. The importance of *K. pneumoniae* capsule in virulence has made it attractive vaccine antigens [34, 35], and therapeutic targets against invasive *Klebsiella* infections [36–38].

The liver is a major organ to trap invading microbes in the blood circulation [39–44]. The liver-resident macrophage Kupffer cells (KCs) has been recognized as the major phagocyte in the sinusoids to fulfill the hepatic anti-infection function [45]. KCs represent approximately 90% of all resident macrophages in the body. Several receptor-mediated pathogen recognition mechanisms are known for KC's capture of blood-borne bacteria, including complement-dependent and -independent means. CRIg, a complement receptor uniquely expressed on KCs [46], is important for complement-mediated bacterial clearance [46, 47]. The von Willebrand factor is shown to recruit platelets to encase bacteria that are bound to KCs [48]. An uncharacterized scavenger receptor(s) of KCs is involved in hepatic recognition of *Listeria monocytogenes* [49]. KCs have been shown to play a critical role in the control of *K. pneumoniae* infection [50, 51], but it remains unclear how KCs interact with the *K. pneumoniae* capsule.

In this study, we found remarkable dominance of several capsule types in causing lethality in a mouse sepsis model among 86 *K. pneumoniae* strains. Capsule types are further shown to dictate the virulence level of *K. pneumoniae* using isogenic derivatives of strain ATCC43816 (K2). We've uncovered that the molecular interactions between the capsule and KCs are primarily responsible for the low-virulence (LV) and high-virulence (HV) phenotypes of *K. pneumoniae*. Vaccination was finally explored to activate the anti-bacterial potential of KCs against HV *K. pneumoniae*.

## Results

### Capsule type defines the virulence level of *K. pneumoniae*

To understand how capsule contributes to *K. pneumoniae* disease potential, we first assessed the virulence level of 81 invasive isolates from blood and other sterile body sites (e.g., bile, bone marrow, cerebrospinal fluid, joint fluid, midstream urine and pus) and 5 laboratory strains in a mouse sepsis model (**S1 Table**). These strains represented 25 capsule types and 35 MLST types. Virulence was characterized by mortality rate in 7 days post intraperitoneal (i.p.) inoculation. This trial showed striking variations in virulence among these strains. For the sake of result analysis, we divided these strains into the categories of high-virulence (HV, 40 strains) and low-virulence (LV, 46 strains), with a cut-off value of 50% mortality within 7 days for the HV strains (**Fig 1A** and **S1 Table**). As represented by **Fig 1B**, infection with HV isolates typically led to 100% mortality in 12 to 48 hr, whereas mice infected by the LV counterparts mostly survived (**Fig 1D**). The virulence phenotypes were consistent with bacteremia kinetics (**Fig 1C** and **1E**). The HV strains usually showed higher levels of bacteremia at 12 and 24 hr as high as $10^9$ colony forming units (CFU)/ml blood. By comparison, the LV strains were not or marginally detectable in the blood. This result showed a wide range of disease potential for *K. pneumoniae* strains. Because the hypermucoviscosity is considered as a virulence determinant of *K. pneumoniae* [8–10], we also characterized the trait by string test of colonies. While many HV strains produced colonies with hypermucoviscosity (31/40, 77.5%), a number of them did not share the phenotype (9/40, 22.5%) (**S1 Table**). In a similar pattern, some of the LV isolates also possess the hypermucoviscosity trait (11/46, 23.9%). This screen did not reveal a clear association between colony hypermucoviscosity and virulence phenotypes.

Further analysis revealed that virulence phenotypes are associated with capsule type (**Fig 1A**). All of the K1, K2, K16 and K20 strains showed the HV phenotype; the remaining types all behaved as the LV strains, except for type 47 that contained a mixture of 2 HV and 8 LV

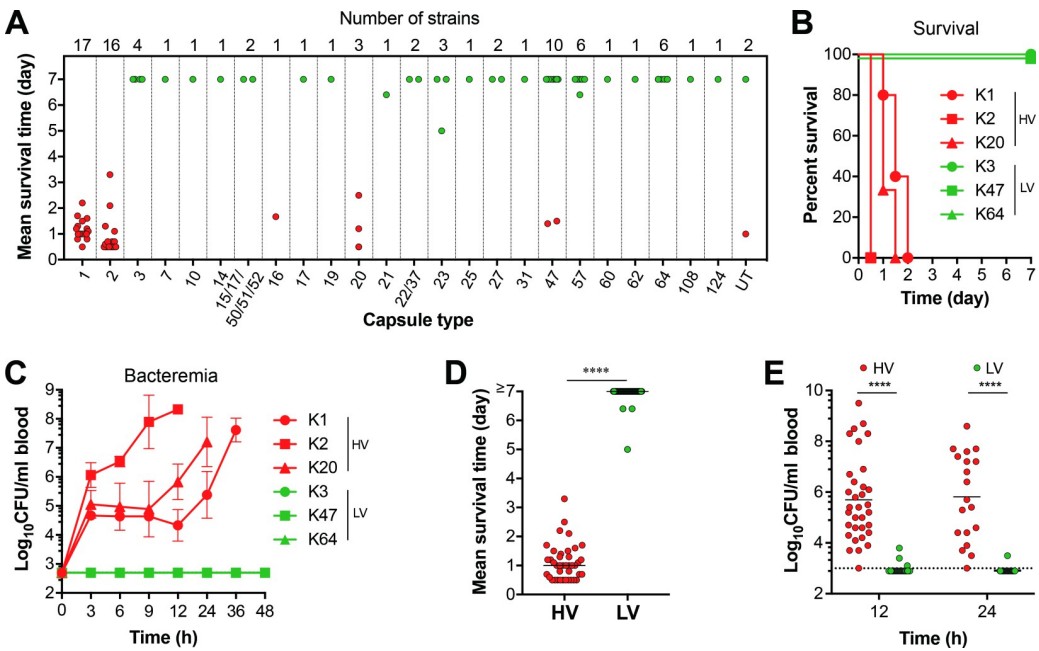

**Fig 1. Dramatic variation in the virulence of *K. pneumoniae* isolates.** Virulence levels of *K. pneumoniae* isolates were assessed in CD1 mice by intraperitoneal (i.p.) infection with $10^6$ CFU of each isolate. **A.** Correlation between capsule type and mean survival time of mice. Mean survival time of mice infected by each high- (HV, red dot) or low-virulent (LV, green dot) isolates in day is presented as a filled circle and grouped in a single column based on capsule type. Capsule type and total isolates tested for each type are indicated at the bottom and top of each group, respectively. UT, untypable. n = 3. **B, C.** Virulence traits of the HV and LV isolates. Survival rate (B) and bacteremia level (C) of representative HV (K1, K2 and K20) and LV (K3, K47 and K64) strains were determined. Strains are denoted by their capsule types: K1 (TH9885), K2 (ATCC43816), K3 (ATCC13883), K20 (TH12908), K47 (TH12846) and K64 (ATCC35657). n = 5–6. The mice died immediately after bleeding at 12 h in (C) were considered as being survived at this time point. The data were pooled from two independent experiments and presented as mean value ± standard deviation (SD). **D, E.** Mean survival time (D) and bacteremia level (E) of the mice at 12 and 24 hr post infection with HV and LV strains. n = 3. Each dot represents the mean value of three mice infected with a single strain. Dash line indicates the detection limit. Unpaired *t* test (D) and two-way ANOVA with Sidak's multiple comparisons test (E) were performed. ****, $P < 0.0001$.

strains. To test if capsule type alone can define virulence phenotype, we constructed isogenic capsule switched derivatives of the HV strain ATCC43816 (K2) [52], in which the capsule biosynthesis (*cps*) gene cluster was replaced by those of donor strains (**Figs 2A** and **S1A**). We chose three donor strains to represent the HV (TH12908, K1) and LV (ATCC13883, K3; TH12852, K23) phenotypes. All the three strains produced colonies without hypermucoviscosity as determined by string test (**S1 Table**). Although the *cps* mutant (Δ*cps*) was avirulent, complementation with the donor *cps* cluster of K1 (K2^K1) or K2 (K2^K2) restored the HV phenotype with 100% mortality in 48 hr (**Fig 2B**) and persistent bacteremia (**Fig 2C**). In sharp contrast, the *cps* gene clusters of K3 (K2^K3) and K23 (K2^K23) failed to resurrect the HV phenotype of the mutant. All of the mice infected with K2^K3 or K2^K23 survived without detectable bacteremia. Since the amount of capsule has been shown to affect the *K. pneumoniae* virulence [53, 54], we quantified capsular polysaccharides (CPS) and mucoviscosity of the capsule switched variants and related strains by uronic acid (UA) and sedimentation assay, respectively. The result revealed certain variations in the CPSs among the *K. pneumoniae* strains, but the amounts of capsules were not correlated with the virulence phenotypes (**Fig 2D**). Among the four selected wild type strains, K1 (HV phenotype) and K3 (LV phenotype) exhibited the highest and lowest CPS production, respectively, while K2 (HV phenotype) and K23 (LV phenotype) produced a similar level of CPSs, despite of their opposite virulence phenotypes. Isogenic K2 hybrid strains

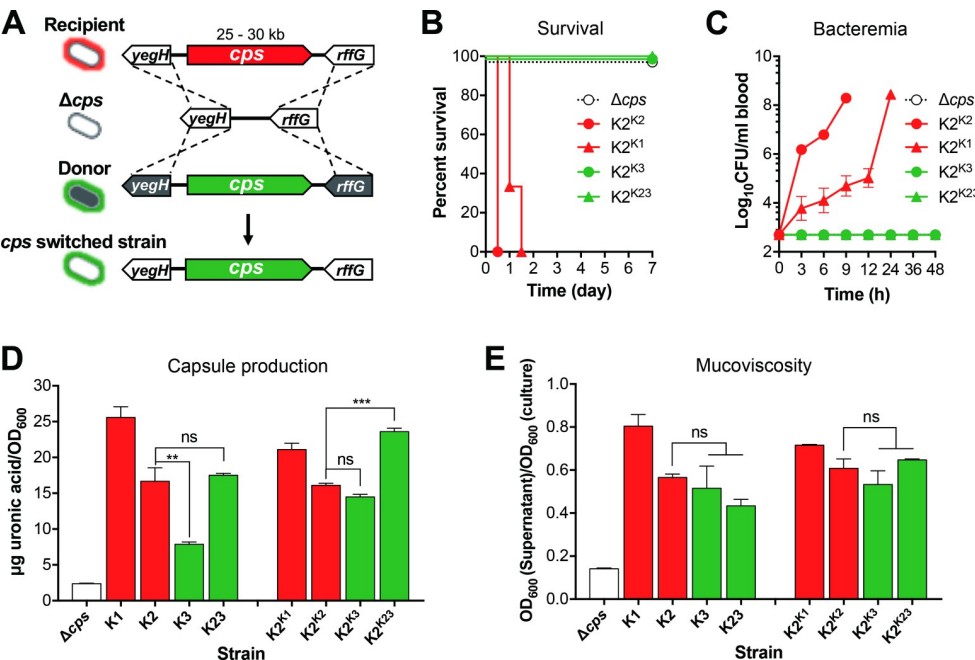

**Fig 2. The role of capsule type in *K. pneumoniae* virulence. A.** Schematic diagram of the capsule type replacement strategy in ATCC43816 (capsule type K2). The *cps* gene cluster of recipient strain (red region) was first deleted to produce an unencapsulated (Δ*cps*) mutant before the *cps* gene amplicon (green region) of donor strain were transformed into the Δ*cps* strain to yield capsule replacement strain. **B, C.** Virulence traits of K2 derivatives. CD1 mice were i.p. infected with $10^6$ CFU of acapsular mutant of ATCC43816 (Δ*cps*) or isogenic derivative carrying the *cps* gene cluster of capsule type K1 from strain TH12908 (K2$^{K1}$), K2 from ATCC43816 (K2$^{K2}$), K3 from ATCC13883 (K2$^{K3}$), and K23 from TH12852 (K2$^{K23}$) to determine survival rate (**B**) and bacteremia level (**C**). Dash line denotes the detection limit. n = 6. **D, E.** Quantification of capsule production (**D**) and mucoviscosity (**E**) of the recipient strain (K2), acapsular mutant (Δ*cps*), capsule switched variants (hybrids) and respective *cps* donor strains (donor). The capsular polysaccharide (CPS) was quantified by the uronic acid method and presented as the amount of uronic acid per OD$_{600}$. Mucoviscosity status was evaluated by the sedimentation assay and shown as the OD$_{600}$ ratio of the supernatant and culture. The data were pooled from two independent experiments and presented as mean ± SD, and statistically tested with one-way ANOVA with Tukey's multiple comparisons. ns, not significant; **, $P < 0.01$; ***, $P < 0.001$.

also displayed significant difference in the amount of CPSs with a high-to-low order of K2$^{K23}$, K2$^{K1}$, and K2$^{K3}$, suggesting the differential impact of genetic background on the production of various capsule types. However, the K23 hybrid (K2$^{K23}$) still retained the low-virulence phenotype as the donor strain (K23). The result shows that capsule type defines *K. pneumoniae* virulence phenotype.

In the context of virulence phenotypes, we compared the isogenic K2 derivatives in colony mucoviscosity. The string test result showed the same mucoviscosity phenotypes between the K2 hybrid strains of K1 (K2$^{K1}$), K2 (K2$^{K2}$) and K3 (K2$^{K3}$), and the donor strains/capsule types (**Table 1**). Specifically, complementation of Δ*cps* with the *cps* genes of the hypermucoviscous K2 strain yielded K2$^{K2}$ hybrid with hypermucoviscous colonies, whereas the K1 (K2$^{K1}$) and K3 (K2$^{K3}$) retained the non-hypermucoviscosity phenotype of the donor strains/type. Surprisingly, transformation of Δ*cps* with the K23 *cps* genes of a non-hypermucoviscosity strain generated the hybrid strain with hypermucoviscous colonies. This result suggests that hypermucoviscosity represents a collective trait of genetic background and capsule type. Quantification of mucoviscosity by centrifugation revealed significant variations among the *K. pneumoniae* strains, but the data did not full recapitulate their string test levels or virulence phenotypes (**Fig 2E**). The K2 and K23 wild type and hybrid strains showed the same

**Table 1. Characteristics of isogenic ATCC43816 derivatives.**

| Strain | String test | Capsule type | Virulence phenotype |
|---|---|---|---|
| K2$^{K1}$ | - | K1 | HV |
| K2$^{K2}$ | + | K2 | HV |
| K2$^{K23}$ | + | K23 | LV |
| K2$^{K3}$ | - | K3 | LV |
| K2$^{K47-H}$ | - | K47 | HV |
| K2$^{K47-L}$ | - | K47 | LV |
| TH12908 | - | K1 | HV |
| ATCC43816 | + | K2 | HV |
| ATCC13883 | - | K3 | LV |
| TH12852 | - | K23 | LV |

mucoviscosity value, respectively, although the capsule types showed the opposite virulence phenotypes (**Fig 2B**). Taken together, our data show that capsule type is the major determinant of *K. pneumoniae* virulence in mice.

## The liver is the major target organ of *K. pneumoniae* capsule for immune evasion

Based on the current understanding that capsules mainly promote bacterial evasion of host defense in the blood circulation during septic infections [12], we compared the dynamics of the isogenic HV and LV strains in the blood of mice post intravenous (i.v.) infection. In agreement with the i.p. infection result (**Fig 2**), the K2$^{K1}$ and K2$^{K2}$ strains showed the HV phenotype with 100% mortality of i.v. infected mice, but all of the mice infected with the K2$^{K3}$ and K2$^{K23}$ strains or acapsular mutant (Δ*cps*) survived (**S2A Fig**). The virulence phenotypes of the HV and LV strains were mirrored by the patterns of bacterial survival in the blood circulation. While the LV K2$^{K3}$ and K2$^{K23}$ strains became undetectable in the first 60 min and remained undetectable ever since, the K2$^{K1}$ and K2$^{K2}$ counterparts showed a transient reduction in bacteremia in the first 3 hr, and recovered later (**S2B Fig**). Further sampling with shorter intervals revealed striking differences in bacterial fate within the first 30 min of septic infection (**Fig 3A**). Acapsular and LV strains K2$^{K3}$, K2$^{K23}$ became undetectable in the bloodstream in the first 20 min, the bacteremia levels of K2$^{K1}$ and K2$^{K2}$ were marginally reduced. This result indicates that the capsule type-dependent fates of *K. pneumoniae* or virulence phenotypes in septic infection are decided by differential clearance of blood-borne bacteria at the very early phase.

To determine how the Δ*cps* and LV strains quickly disappeared from the circulation, we determined bacterial distribution in five major organs (heart, kidney, liver, lung, and spleen) in the first 30 min post infection. The total bacteria of Δ*cps* in mice was significantly reduced by 41.4% as early as 5 min post infection, compared to the initial inoculum, whereas the bacterial load of the encapsulated variants did not change significantly (**S2C Fig**). The vast majority of the viable Δ*cps* mutant (92.1%) and LV encapsulated strains (K2$^{K3}$, 84.5%; K2$^{K23}$, 99.1%) were found in the livers as early as 5 min, whereas the HV counterparts predominantly remained in the blood (K2$^{K1}$, 62.7%; K2$^{K2}$, 77.4%) and were marginally detected in the liver (K2$^{K1}$,13.5%; K2$^{K2}$, 3.9%) (**Fig 3B** and **S2 Table**). The spleen was the second organ with the high bacterial burden for Δ*cps* (4.6%), K2$^{K3}$ (12.7%) and K2$^{K23}$ (0.7%) strains. K2$^{K1}$ (16.8%) and K2$^{K2}$ (12.0%) were relatively more abundant in the spleen than the liver. This capsule type-dependent hepatic sequestration of *K. pneumoniae* was also observed at 30 min (**Fig 3C**). The liver was still the major organ with the highest bacterial burden for Δ*cps* and LV

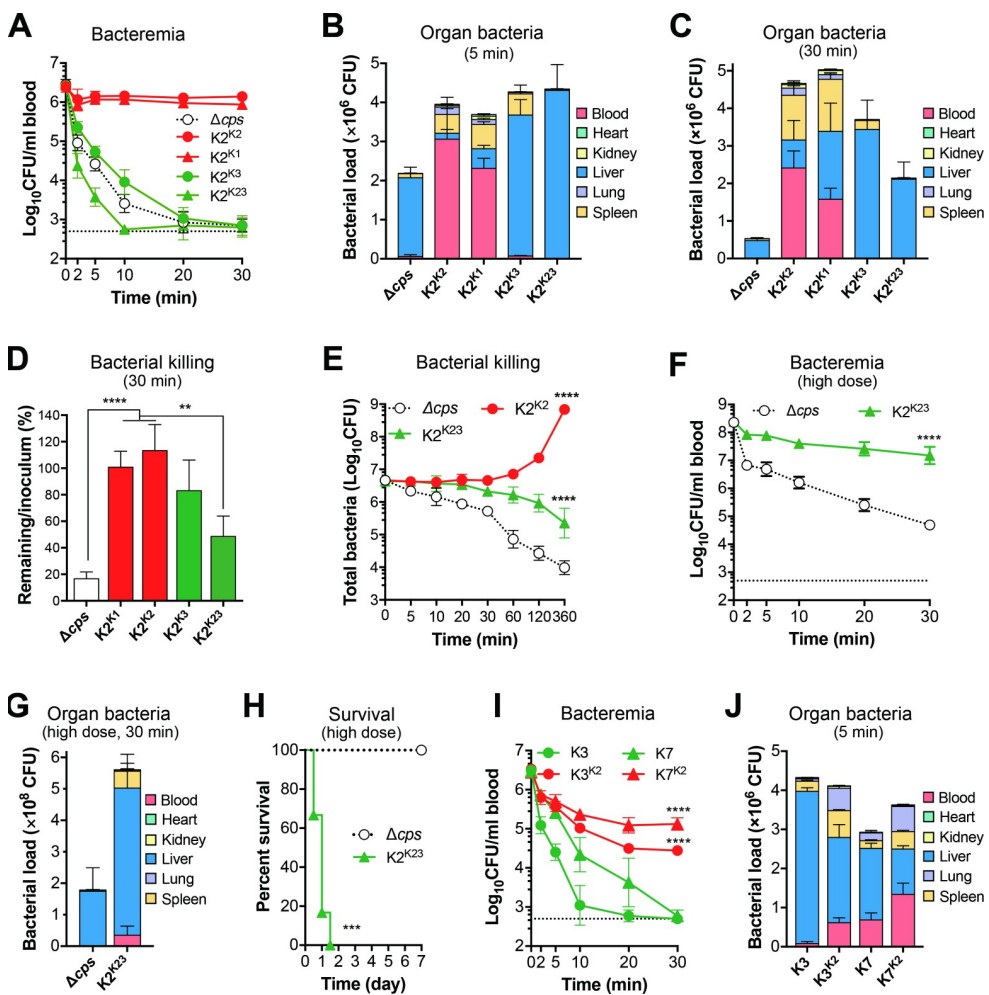

**Fig 3. Capsule type-dependent capture of circulating *K. pneumoniae* in the liver. A-D.** Superior protection of HV capsule types against hepatic clearance of *K. pneumoniae* in the early phase of septic infection. ATCC43816 (K2) derivatives in the first 30 min of septic infection. CD1 mice were intravenously (i.v.) infected with $5 \times 10^6$ CFU of isogenic acapsular mutant ($\Delta cps$), and capsule switched HV ($K2^{K1}$ and $K2^{K2}$) or LV ($K2^{K3}$ and $K2^{K23}$) derivatives. CFU counts in the blood at various time points (**A**) and the major organs at 5 min (**B**) and 30 min (**C**), were determined; bacterial killing in the first 30 min was estimated by comparing the combined CFU detected in the blood and five organs with the corresponding inoculum (**D**). n = 6. **E.** The impact of the capsule and capsule types on bacterial survival/killing in the first 6 hr of septic infection. CD1 mice were infected with the capsular mutant ($\Delta cps$), capsule switched HV derivative ($K2^{K2}$) or LV derivative ($K2^{K23}$) as in (A); the CFU counts obtained from the blood and five organs of each mouse are combined and presented as a single value at each time point. n = 6. **F-H.** The contribution of LV capsule type to bacterial survival. CD1 mice were i.v. infected with $5 \times 10^8$ CFU of acapsular ($\Delta cps$) or LV ($K2^{K23}$) derivative of ATCC43816; CFU counts obtained with the blood (**F**) and major organs (**G**) samples; mouse survival observed for 7 days post infection (**H**). n = 6. **I, J.** K2 capsule-mediated protection of K3 and K7 strains against hepatic capture. LV strains K3 (TH12849) and K7 (TH12880) were compared with their progenitors K2 capsule-switched derivatives ($K3^{K2}$ and $K7^{K2}$) in CFU counts in the blood (**I**) and major organs (**J**) as in (A) and (B), respectively. n = 4. The data were pooled from two independent experiments in CD1 mice and presented as mean ± SD, except for data of (H), which were from one experiment. One-way ANOVA with Tukey's multiple comparisons test (D), two-way ANOVA with Tukey's multiple comparisons test (E, F and I) and log-rank test (H) were performed. **, $P < 0.01$; ***, $P < 0.001$; ****, $P < 0.0001$.

derivatives, and the blood and spleen held the majority of HV strains at this time point. We further verified the capsule type-dependent bacterial capture by testing a representative clinical isolate for each of the 21 additional capsule types. In line with the virulence phenotypes (**S1 Table**), K16 and K20 strains showed similarly strong resistance to hepatic capture as K1 and

K2 counterparts. By comparison, all of the other 19 LV capsule types were highly susceptible to hepatic trapping (S2D Fig).

Comparative analysis of the remaining bacteria in mice at various time points post infection indicated that the bacteria cleared from the bloodstream are effectively killed after being captured by the liver and perhaps other organs. As compared with the inoculum, the total CFU counts of isogenic $\Delta cps$ and LV variants in the blood and five organs were significantly reduced at 30 min, with $\Delta cps$ being the most dramatically decreased (Fig 3D). In contrast, the total CFU counts of HV variants $K2^{K1}$ and $K2^{K2}$ were maintained at the level of the inoculum ($K2^{K1}$) or slightly increased ($K2^{K2}$). This result indicated that the bacteria captured in the liver are killed in the first 30 min of septic infection. It is worth noting that $K2^{K23}$ bacteria were removed from the body at a faster pace than $K2^{K3}$ (Fig 3A), suggesting that capsule types also affect bacterial killing capacity of the host (S2E–S2J Fig).

Since the LV variants were still abundantly detected in mice at 30 min post infection, we compared the total CFU counts of LV $K2^{K23}$ strain with those of the isogenic acapsular and HV representative strains in mice at various time points in 6 hr post i.v. infection. Consistent with the potent survival in the blood circulation in the early time points, the HV $K2^{K2}$ showed steady growth starting at 60 min; the total CFU counts were increased by 152.8 fold at 6 hr (Fig 3E). By comparison, the LV $K2^{K23}$ displayed a gradual reduction in the total bacteria, with an overall reduction by 10.0 fold at 6 hr, as compared with the inoculum. Acapsular $\Delta cps$ disappeared with the fastest pace in this period, with much lower levels of over bacterial load than the LV $K2^{K23}$ in mice at 60 min (by 22.1 fold), 120 (by 36.3 fold) and 360 min (by 29.8 fold). This result has uncovered that the LV capsule type still confers certain resistance to host bactericidal immunity even after being captured in the liver. To assess the contribution of the LV capsule types to *K. pneumoniae* virulence, we directly compared the clearance of isogenic acapsular and $K2^{K23}$ strains at a higher infection dose ($5 \times 10^8$ CFU). The LV strain showed stronger resistance to host clearance from the bloodstream (Fig 3F) to the liver (Fig 3G). Accordingly, the LV bacteria has significant advantage over acapsular bacteria in the overall viability in mice (Fig 3G) and lethality (Fig 3H). These data show that the *K. pneumoniae* capsule promotes bacterial survival and virulence by escaping hepatic capture and killing in a capsule type-dependent manner. The "avirulent" status of $\Delta cps$ justifies our designation of certain *K. pneumoniae* capsule types as "low-virulence" types.

To verify the dominant role of capsule type in evading bacterial capture in the liver, we genetically replaced the *cps* genes of the LV strains TH12849 (capsule type K3) and (capsule type K7) with those of the HV ATCC43816 (K2). The K2 capsule-producing variants of the K3 ($K3^{K2}$) and K7 ($K7^{K2}$) strains showed significantly slower clearance from the bloodstream than the LV recipients K3 and K7 (Fig 3I). As an example, the bacterial levels of $K3^{K2}$ and $K7^{K2}$ were higher than the parental K3 (by 55.5 fold) and K7 (by 223.0 fold) strains at 30 min. Further tracking of bacteria in major organs revealed that the K2 capsule variants were more resistant to hepatic capture than the parental strains (Fig 3J). The livers of mice infected by $K3^{K2}$ and $K7^{K2}$ contained 53.0% and 31.9% of total detectable bacteria, respectively at 5 min, which were significantly lower than those of the parental strains (K3, 89.9%; K7, 62.4%). A similar trend of K2 capsule-mediated immune evasion was also observed at 30 min after infection (S2K Fig). These results showed the K2 capsule significantly enables *K. pneumoniae* to resist host clearance from the bloodstream.

Comparison analysis of the three K2 capsule-producing variants revealed significant impact of strain backgrounds on the immune evasion of the K2 capsule. All mice survived i.v. infection with $5 \times 10^6$ CFU of $K3^{K2}$ or $K7^{K2}$ (S2L Fig), whereas the same dose of $K2^{K2}$ yielded 100% mortality in 24 hr post infection (S2A Fig). Accordingly, mice infected with $K2^{K2}$ showed a higher level of bacteremia than $K3^{K2}$ (by 48.5 fold) and $K7^{K2}$ (9.6 fold) at 30 min post infection

(**Fig 3A and 3I**). To assess potential impact of CPS level on the function of K2 capsule, we quantified the CPSs of the isogenic K2 capsule-producing strains. As compared with $K2^{K2}$, $K3^{K2}$ produced less CPS, while $K7^{K2}$ showed an equivalent level of the capsule (**S2M Fig**), indicating that the abundance of the K2 CPS in these strains is not responsible for the differences in the immune of the K2 capsule. We further assessed mucoviscosity of the K2 capsule variants. Surprisingly, in contrast to the hypermucoviscous feature of $K2^{K2}$ colonies, $K3^{K2}$ and $K7^{K2}$ produced hypermucoviscosity-negative colonies as assessed by string test (**S1 Table**), which is consistent with the difference of the K2 capsule variants in virulence phenotype. In summary, these results indicate that the immune evasion of the high-virulence capsules can be significantly affected by the genetic backgrounds or non-*cps* genes of *K. pneumoniae*.

## The spleen is dispensable for the clearance of low-virulence *K. pneumoniae* in the early phase of septic infection

The spleen is regarded as an important immune organ for combating invasive infections, particularly by encapsulated bacteria [55]. In the context of the dominant role of the liver in capturing the LV capsule types of *K. pneumoniae*, we evaluated the contribution of the spleen in the clearance of *K. pneumoniae* trapping using mice with splenectomy (SPX) or sham operation (SHM). Surgical removal of the spleens did not yield significant impact on the clearance of isogenic blood-borne HV ($K2^{K2}$), LV ($K2^{K23}$) or acapsular ($\Delta cps$) strains (**Fig 4A**). As compared with the control mice, the asplenic mice showed a modest increase of *K. pneumoniae* $K2^{K2}$ in the bloodstream (by 35.7% or 0.4 fold) and lungs (by 46.1% or 0.5 fold) at 30 min post infection (**Fig 4B**). The spillover of HV bacteria to the blood and other organs indicates that the spleen contributes to the clearance of blood-borne HV *K. pneumoniae* although the splenic clearance of HV *K. pneumoniae* was much less effective than the hepatic clearance of LV bacteria. These findings show that the spleen is partially capable of intercepting HV *K. pneumoniae* but dispensable for the capture of acapsular and LV bacteria when the antibacterial function of the liver is intact.

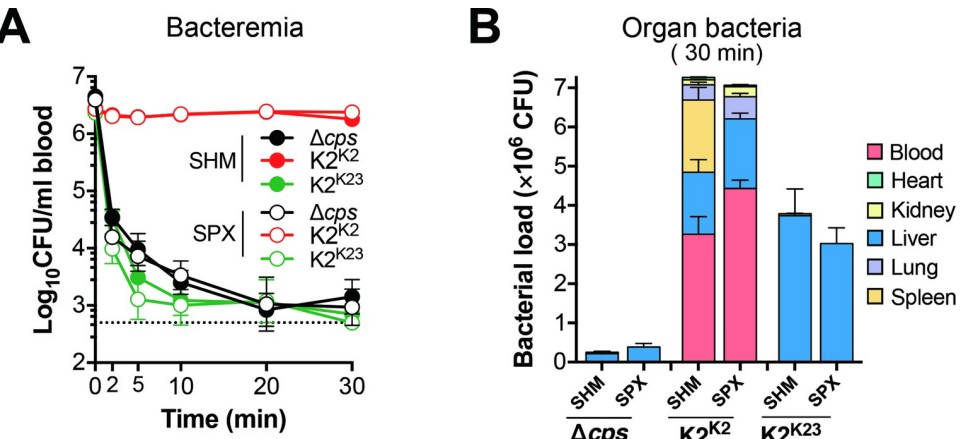

**Fig 4. Dispensable role of the spleen in the clearance of blood-borne *K. pneumoniae* in the early phase of septic infection. A.** Bacterial kinetics in the blood circulation of asplenic mice. The spleens of CD1 mice were surgically removed (splenectomy, SPX) 7 days before i.v. infection with $5 \times 10^6$ CFU of ATCC43816 (K2) or isogenic acapsular mutant ($\Delta cps$) and capsule switched LV strain ($K2^{K23}$). CFU counts in the blood were enumerated on agar plates. Mice with sham operation (SHM) were used as a negative control. Dash line denotes the detection limit. n = 4. **B.** Bacterial distribution in the major organs of asplenic mice. CFU counts in the blood and major organs of asplenic mice were compared with those of sham operation control 30 min post i.v. infection as in (A). n = 4. The data were from one experiment and presented as mean ± SD.

## Capsule type dictates the capacity of Kupffer cells in capturing *K. pneumoniae* in the liver

Macrophages, monocytes and neutrophils have been shown to control *K. pneumoniae* infection under various *in vitro* and *in vivo* conditions [1], but it is unknown how the innate immune systems in the liver differentiate the LV and HV capsule types of *K. pneumoniae*. We first assessed the contribution of neutrophils and monocytes to the clearance of K2$^{K23}$ *K. pneumoniae* by selective depletion with 1A8 (targeting neutrophil) or Gr1 (depleting neutrophil and inflammatory monocyte) antibodies, and the efficiency of immune cell depletion was determined (S3 Fig). This treatment did not affect the clearance rate (Fig 5A) and hepatic capture (Fig 5B) in the first 30 min. On the contrary, depletion of macrophages with clodronate-containing liposomes (CLL) dramatically impaired bacterial clearance at various time points at the beginning of septic infection (Fig 5A). Consistently, hepatic trapping of K2$^{K23}$ *K. pneumoniae* from the blood circulation was reduced by 61.2% (Fig 5B). Instead, significantly more bacteria were found in the blood (by 34.2%) and spleen (by 26.9%) of CLL-treated mice as compared with the control. This result strongly suggests that macrophages but not neutrophils or monocytes are responsible for the capture of circulating LV *K. pneumoniae* in the liver.

Since liver resident macrophage Kupffer cells (KCs) account for 80–90% macrophages in the body [56], we determined the role of KCs in the capture of *K. pneumoniae* by specific depletion in the *Clec4f*-DTR mice with diphtheria toxin (DT) [57]. *Clec4f*-DTR mice specifically expressed the human diphtheria toxin receptor (DTR) that was transcriptionally driven by the promoter of the KC-specific *Clec4f* gene, which made DT toxicity/depletion to KC primarily. Depletion of KCs dramatically reduced the clearance of acapsular and K2$^{K23}$ *K. pneumoniae* from the bloodstream (Fig 5C). As an example, the bacteria trapped in the liver were reduced from 87.2% (-DT) to 11.0% (+DT) for Δ*cps*, and from 98.6% (-DT) to 15.9% for K2$^{K23}$ (+DT) at 30 min (Fig 5D). As expected, the depletion of KCs made mice susceptible to LV K2$^{K23}$ variant (Fig 5E).

Hepatic capture of *K. pneumoniae* was further verified by intravital microscopy (IVM) imaging of the liver sinusoids. The Δ*cps* and K2$^{K23}$ strains were rapidly captured by KCs, while K2$^{K2}$ bacteria predominantly flowed through the sinusoidal vasculatures (Fig 5F and S1 Movie). The Δ*cps* and K2$^{K23}$ per field of view (FOV) were 8.4 and 6.3 fold respectively higher than that of K2$^{K2}$ (Fig 5F, right panel). In agreement with sparse distribution of KCs in the liver sinusoids of DT-treated *Clec4f*-DTR mice, the mice virtually lost the ability to capture *K. pneumoniae*; the vast majority of the Δ*cps* and LV bacteria moved rapidly through vasculatures (Fig 5G, right panel; S2 and S3 Movies). With the depletion of KCs, the number of Δ*cps* and LV bacteria in liver per FOV reduced from 90 to 6 (15.0 fold) and 76 to 16 (4.8 fold), respectively (Fig 5G, left panel). To determine whether bacteria captured by KCs are internalized by phagocytosis, we assessed bacterial uptake by differentiating extracellular from intracellular bacteria by IVM using pH-sensitive FITC (green fluorescence at the extracellular neutral pH) and pHrodo (red at the phagolysosomal acidic pH) dyes [58] after i.v. infection with *K. pneumoniae* double labeled with FITC and pHrodo. Certain pHrodo-labeled bacteria on KCs became red from 5 to 30 min after infection, indicative of active bacterial phagocytosis by KCs (Fig 5H). Quantitative data revealed that KCs took up substantial levels of the captured Δ*cps* (27.4% per FOV) and K2$^{K23}$ (11.5% per FOV) bacteria in the first 30 min post infection. The IVM experiments also indicate that KCs are more capable of taking up acapsular than encapsulated LV *K. pneumoniae* by two consecutive steps: capture (binding) and phagocytosis (killing). These results have uncovered that the capsule mainly promote *K. pneumoniae* survival in the bloodstream by evading the capture of KCs in the liver sinusoids in a capsule type-dependent manner, which is consistent with the drastic difference of the LV and HV capsule types in virulence phenotypes.

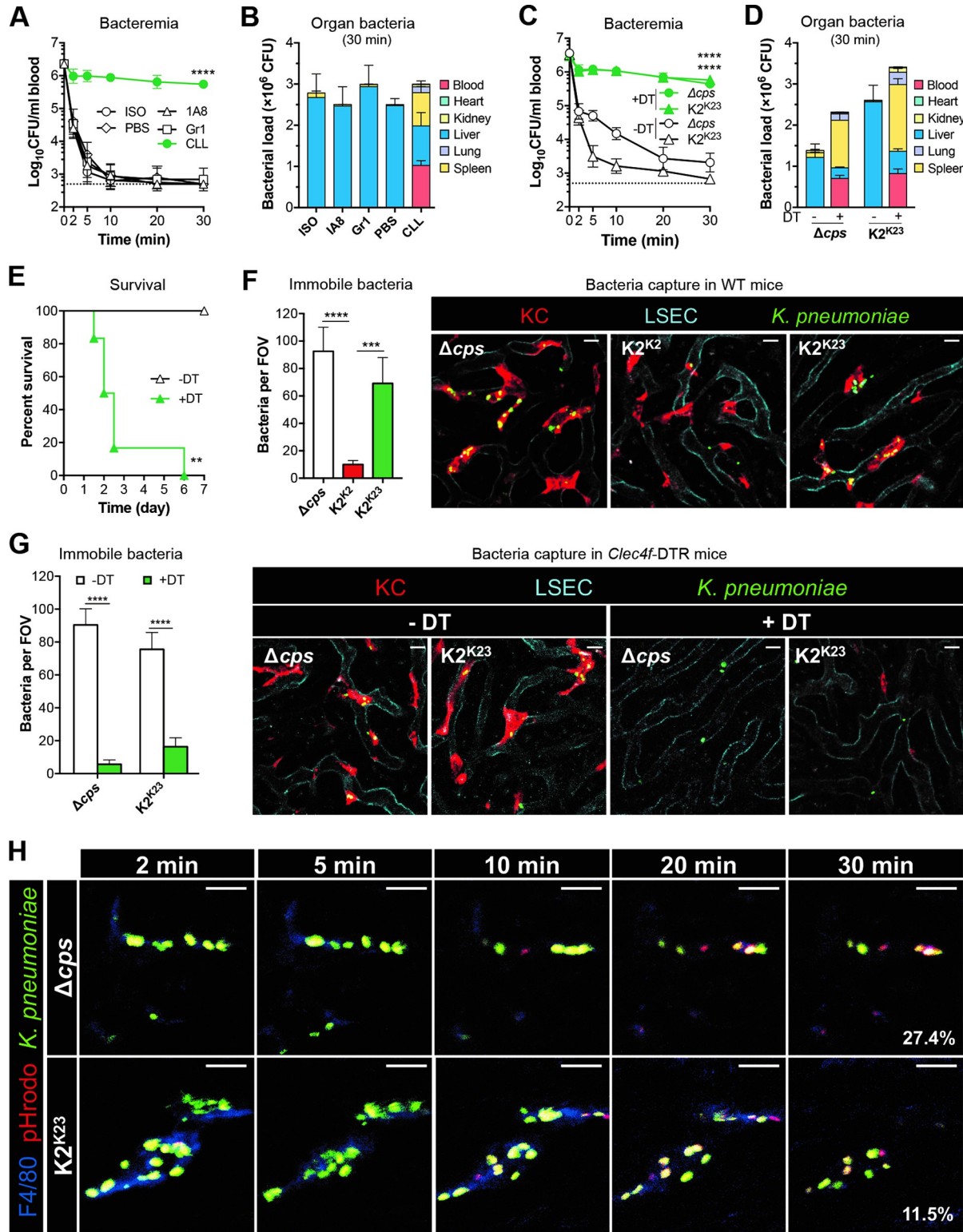

**Fig 5. Capsule type-dependent capture of *K. pneumoniae* by KCs. A, B.** Importance of macrophages in clearing the LV derivative (K2^K23) of ATCC43816. CD1 mice were pretreated with PBS, ISO (isotype control), CLL (depleting macrophage), 1A8 antibody (depleting neutrophil) or Gr1 antibody (depleting neutrophil and inflammatory monocyte) before being i.v. infected with $5 \times 10^6$ CFU of K2^K23 to determine bacteria in the bloodstream (**A**) and organs (**B**). n = 6. **C, D.** Essentiality of KCs in capturing acapsular and LV variants of ATCC43816. *Clec4f*-DTR mice were pretreated with (+DT) or without (-DT) diphtheria toxin and i.v. infected with $5 \times 10^6$ CFU of the

acapsular ($\Delta cps$) and LV derivative (K2$^{K23}$) of ATCC43816 to determine bacteria in the bloodstream (**C**) and organs (**D**). Dash line denotes the detection limit. n = 6. **E.** The importance of KCs in the host survival against infection of LV *K. pneumoniae*. *Clec4f*-DTR mice were pretreated with (+DT) or without (-DT) diphtheria toxin and i.v. infected with $2 \times 10^7$ CFU of K2$^{K23}$. n = 6. **F.** Intravital microscopy (IVM) detection of *K. pneumoniae* capture by KCs in the liver sinusoids. CD1 mice were pretreated with antibodies to fluorescently labelling KC (red), liver sinusoidal endothelial cells (LSEC) (cyan), i.v infected with $5 \times 10^7$ CFU of isogenic HV (K2), LV (K2$^{K23}$) or $\Delta cps$ *K. pneumoniae* (green), and imaged for bacterium-KC interactions in the liver sinusoids approximately 10 min post infection. Immobilized bacteria on KCs and occasionally vascular wall were quantified in 6 random fields of images (right) and presented as bacteria per field of view (FOV) (left). The process of bacterial interaction with KCs is manifested in **S1 Movie**. Scale bar, 10 µm. n = 2. **G.** The loss of *K. pneumoniae* capture in the liver in KC-deficient mice. *Clsec4f*-DTR mice were pretreated with (+DT) or without (-DT) diphtheria toxin before being used to detect KC capture of $\Delta cps$ and K2$^{K23}$ by IVM as in (F). **H.** Phagocytosis of $\Delta cps$ and K2$^{K23}$ by KCs. The phagocytosis was assessed by i.v. infection of mice with bacteria double labeled with FITC and pH-sensitive pHrodo dyes; extracellular (green) and intracellular (red) localization of bacteria bound by KCs (blue) were tracked by IVM within 2–30 min after infection as in (F). The internalized bacteria changed the color appearance from green (FITC), yellow (FITC + pHrodo) to red (pHrodo) after pHrodo was activated in the low pH environment where FITC lost fluorescence. Six random fields of representative images at 30 min were used to quantify the extracellular and intracellular bacteria. The percentage of intracellular bacteria out of the total immobilized bacteria per field is indicated in the images (30 min). Scale bar, 10 µm. The data of (A-E) were pooled from two independent experiments and presented as mean ± SD. Two-way ANOVA with Tukey's (A and C) or Sidak's (G) multiple comparisons test, log-rank test (E) and ordinary one-way ANOVA with Tukey's multiple comparisons test (F) were performed. **, $P < 0.01$; ***, $P < 0.001$; ****, $P < 0.0001$.

## Capsule polysaccharides of LV types are recognized by KCs

Effective capture of encapsulated LV *K. pneumoniae* by indicates that KCs capture the LV *K. pneumoniae* by recognizing CPS of LV capsule types or other cell-surface-exposed microbial ligands that are not fully covered by the LV capsule variants. We differentiated these possibilities by i.v. inoculation of free CPS before bacterial infection, which would temporarily saturate the CPS receptors on KCs and thereby block the macrophages from recognizing the bacteria coated by the same types of capsules. Intravenous injection of K23 CPS (CPS-K23) approximately 2 min before i.v. inoculation of K2$^{K23}$ *K. pneumoniae* led to dose-dependent reduction of bacterial clearance or increased bacteremia (**Fig 6A**). As an example, treatment with 800 µg CPS-K23 elongated the 50% clearance time from 0.3 to 2.4 min (**S4A Fig**). The delayed clearance was also reflected by dose-dependent decline in hepatic-captured bacteria (**Fig 6B**). The inhibitory effect of free CPS on bacterial clearance was also confirmed with purified CPS of K3 (CPS-K3) or KL108 (CPS-K108) before infection with homologous strain (**Fig 6C**). These experiments show that KCs capture the LV *K. pneumoniae* by recognizing the CPS. To determine whether the blocking effects of free CPS is type-specific, the mice were pretreated with 800 µg CPS of K3, K23 or KL108 before i.v. infection with a strain of homologous or heterologous capsule type. The clearance of K2$^{K23}$, K2$^{K3}$ and KL108 was blocked only by CPS-K23, CPS-K3 and CPS-KL108, respectively, but not any CPS of heterologous capsule types (**Fig 6D**). As an example, CPS-K3 only blocked the clearance of K2$^{K3}$ *K. pneumoniae*, but not that of K2$^{K23}$ or KL108, indicating that the recognition of LV strains by KCs is capsule type-specific.

To identify the host factors that mediate KC recognition of *K. pneumoniae* CPS, we first assessed the role of the complement system using mice deficient in complement protein C3 (C3$^{-/-}$) or CRIg (CRIg$^{-/-}$). The C3 activation has been shown to promote complement-mediated killing of *K. pneumoniae* [59–61]; CRIg is the major C3 receptor on KCs [46], and has been reported to mediate hepatic clearance of blood-borne bacterial pathogens [46, 47]. K2$^{K23}$ bacteria were effectively cleared from the bloodstream of wild type, C3$^{-/-}$ and CRIg$^{-/-}$ mice at a similar pace (**Fig 6E**), revealing that the complement system is not involved in the clearance of type K23 *K. pneumoniae*. We next tested potential function of scavenger receptors in the clearance of type K23 *K. pneumoniae* using polyinosinic-acid (poly(I)), a pan-scavenger receptor inhibitor [49]. An uncharacterized scavenger receptor(s) is shown to mediate the pathogen recognition by KCs [49]. Pretreatment of mice with poly(I) significantly delayed the clearance of K2$^{K23}$, as compared with the poly(C) negative control (**Fig 6F**). These findings indicate that

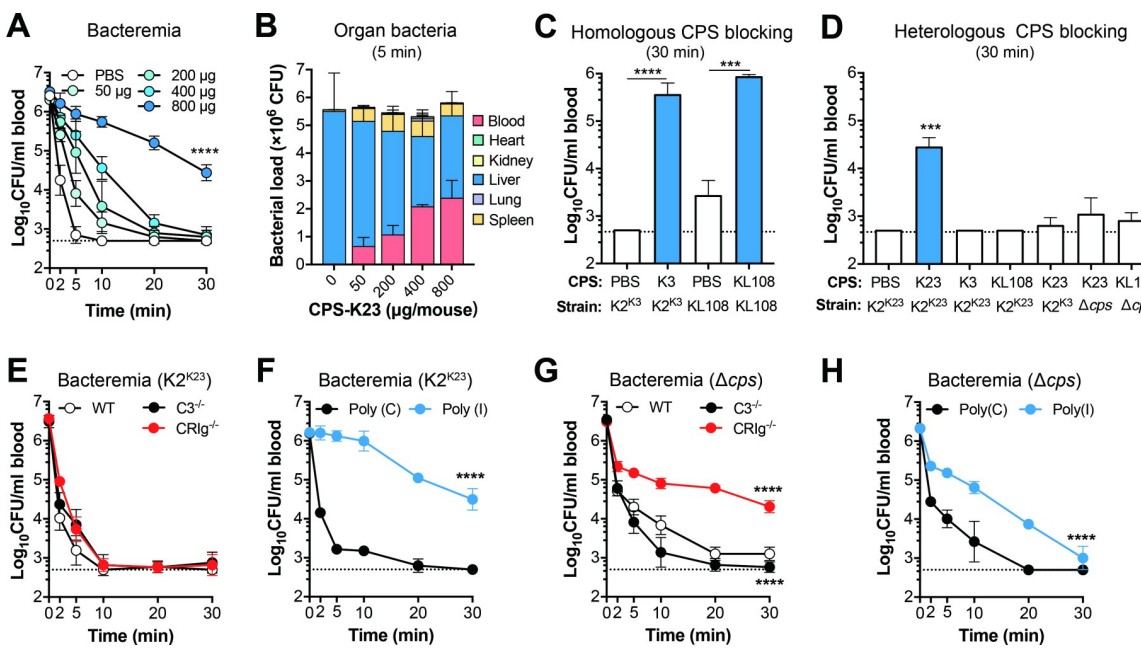

**Fig 6. Receptor-dependent recognition of *K. pneumoniae* in the liver. A.** Dose dependent inhibition of free K23 CPS against the clearance of K2^K23 from the bloodstream. CD1 mice were intravenously inoculated with PBS or various amounts of purified CPS from K2^K23 2 min before i.v. infection with 5 × 10^6 CFU of K2^K23 and determine blood bacteria at 5 time points in the first 30 min of infection. n = 3. **B.** Dose dependent inhibition of free K23 CPS against hepatic capture of K2^K23 in CD1 mice treated as in (**A**). n = 3. **C, D.** Capsule type-specific blocking of free CPS against the clearance of LV variants. CD1 mice were pretreated with PBS or 800 μg free CPS of types K3, K23 or KL108 before infection with the strain of homologous (**C**) or heterologous (**D**) capsule type and enumeration of blood bacteria at 30 min. Capsule type of free CPS and strain inoculated are indicated at the bottom of each column. The strains used are specified in S1 Table. n = 3. **E, F.** Role of the complement system and scavenger receptors in the clearance of blood-borne K2^K23. The involvement of the complement system in the clearance of LV *K. pneumoniae* was assessed by comparing C57BL/6 (WT) mice to those deficient in complement protein C3 (C3^-/-) or major C3 receptor CRIg on KCs (CRIg^-/-) after i.v. infection with 5 × 10^6 CFU of K2^K23 to determine bacteria in the bloodstream in the first 30 min (**E**). The contribution of scavenger receptors in the clearance of LV *K. pneumoniae* was evaluated by comparing CD1 mice with those pretreated with i.v. inoculation of 400 μg poly(I), a pan-scavenger receptor inhibitor, 2 min before infection with K2^K23 and enumeration of bacteria in the bloodstream as in (E). Mice treated with 400 μg poly(C) were used as a negative control. n = 5. **G, H.** Role of the complement system and scavenger receptors in the clearance of blood-borne acapsular *K. pneumoniae*. Same as in (E) and (F) except for using the Δ*cps* strain in i.v. infection. Dash line denotes the detection limit. The data are presented as mean ± SD. Two-way ANOVA with Tukey's multiple comparisons (A, E, F, G and H) and one-way ANOVA test (C and D) were performed. ***, $P < 0.001$; ****, $P < 0.0001$.

KCs recognizes the K23 capsule via an unknown scavenger receptor(s). We tested the impact of the complement system and scavenger receptors on the clearance of Δ*cps* in a similar manner. The deficiency of CRIg, but not C3, significantly impaired the clearance of Δ*cps* strain from the circulation (**Fig 6G**). Likewise, blocking scavenger receptors also led to significant delay in the clearance of acapsular *K. pneumoniae* (**Fig 6H**). The results indicate that CRIg and scavenger receptor(s) mediate the KC recognition of acapsular bacteria.

## Sequence variations in the *cps* locus determine virulence phenotypes of K47 isolates

Our initial screening revealed a clear association between the virulence phenotypes and capsule types, with an exception of capsule type K47 that included 2 HV and 8 LV isolates (**Fig 1A**, **S1 Table**). To understand what is responsible for virulence difference among K47 strains, we chose two representative K47 strains each with a HV (TH12845 or K47-H) or LV (TH12846, K47-L) phenotype in the earlier screen (**S1 Table**). Accordingly, K47-H showed a high-virulence phenotype with 100% mortality of mice within 4 days post i.p. infection,

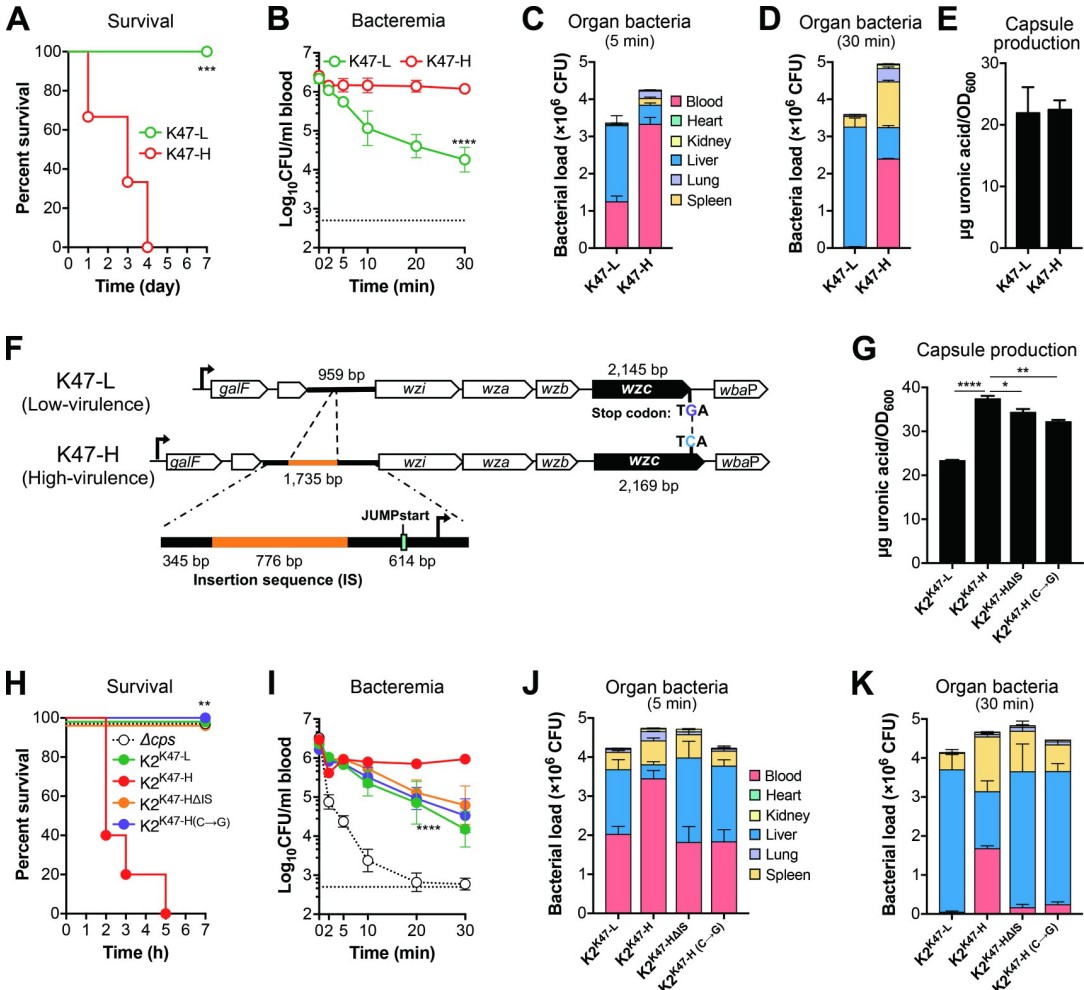

**Fig 7. Variation in virulence among type K47 strains. A.** Virulence of capsule type-K47 HV and LV strains. CD1 mice were i. p. infected with $10^6$ CFU of high-virulence TH12845 (K47-H) or low-virulence strain TH12846 (K47-L). Mortality was assessed in a 7-day period. n = 6. **B-D.** Difference of HV and LV K47 strains in resisting hepatic capture. CD1 mice were i.v. infected with $5 \times 10^6$ CFU of K47-H and K47-L to determine bacteria in the bloodstream (**B**) and major organs at 5 min (**C**) and 30 min (**D**). n = 4. **E.** Capsule production of K47-H and K47-L. K47 CPS was quantified as in Fig 2D. **F.** Schematic illustration of sequence difference in the K47 *cps* locus between K47-H and K47-L. The DNA sequences between the *galF* and *wbaP* genes of the K47 *cps* locus of K47-H and K47-L are drawn to show the two sequence differences between the two strains. The coding sequences are identified with gene names. The 776-bp insertion sequence upstream of *wzi* in K47-H is marked with a yellow line; the C-G SNP at the 2144[th] nucleotide of the *wzc* coding region in K47-L indicated with a vertical dashed line between the two sequences; putative promoter and JUMPstart sequences marked with black arrow and indigo rectangle, respectively. **G.** Capsule production of K47 variants was quantified as in (E). **H.** Effects of the IS and *wzc* SNP on the virulence of K47 variants. Mice were infected with the K47 *cps* variants of ATCC43816 to determine survival rates as in (A). n = 5. **I-K.** Virulence traits of the K47 *cps* variants. Bacteremia kinetics (**I**) and bacterial load in major organs at 5 min (**J**) and 30 min (**K**) were determined in mice i.v. infected as in (B). All data are presented as mean ± SD. Log-rank test (A and G), two-way ANOVA with Tukey's multiple comparisons test (B and I) and one-way ANOVA with Tukey's multiple comparisons test (L) were performed. **, $P < 0.01$; ***, $P < 0.001$; ****, $P < 0.0001$.

whereas all the mice infected with the same dose of K47-L survived (**Fig 7A**). Along the same line, as compare with K47-L, K47-H showed higher levels of survival in the bloodstream (**Fig 7B**) and less capture in the liver (**Fig 7C and 7D**). In contrast to the striking phenotypic difference, both the strains showed a comparable level of capsule production (**Fig 7E**). However, sequencing comparison of the *cps* gene clusters between the strains revealed virtually identical DNA sequence across the entire *cps* locus with differences in only two sites. The first

polymorphism is characterized by the presence of a 776-bp insertion sequence (IS)-like element in the *cps* locus of K47-H 614 base-pair (bp) upstream of the *wzi* gene (**Fig 7F**), which encodes a putative transposase. Bioinformatic analysis identified three promoter-like sequences and a 39-bp JUMPstart element in the vicinity of the insertion site. JUMPstart is frequently located between the promoter and downstream coding sequences of the genes involved in the synthesis of capsules and O antigens in Gram-negative bacteria [62], and has been shown as the binding site for the transcriptional anti-terminator RfaH to promote transcriptional read-through of the downstream genes [63–65]. The other difference is a C-G single nucleotide polymorphism (SNP) at the 2144th nucleotide of the *wzc* coding region in K47-L, which results in premature stop of *wzc* and loss of amino acids at the C terminus of Wzc, a tyrosine autokinase that regulates the cross-membrane channel for CPS export [16, 66].

To ascertain if the two sequence differences are responsible for the phenotypic differences between K47-H and K47-L in septic infection, we made multiple unsuccessful attempts to generate individual mutations in the genomes of the two strains due to their poor transformation. We then characterized their *cps* genes of K47-H and K47-L in the K2 strain ATCC43816. The capsule switched variant $K2^{K47-H}$ produced a significantly higher level of the capsule than $K2^{K47-L}$ (**Fig 7G**), which is in contrast with a similar level of capsule production between K47-H and K47-L (**Fig 7E**). Furthermore, the hybrid strains recapitulated the HV ($K2^{K47-H}$) and LV ($K2^{K47-L}$) phenotypes of the donor strains (**Fig 7H**). Infection with $K2^{K47-H}$ led to 100% mortality of mice in 5 days, whereas all of the mice infected with $K2^{K47-L}$ survived. Furthermore, introducing the C-G *wzc* mutation or removing the IS sequence in the *cps* cluster of K47-H ($K2^{K47-H(C\rightarrow G)}$) led to significant reduction in capsule production although they still produced more CPS than $K2^{K47-L}$ (**Fig 7G**). The C-G mutation in the K47-H *cps* cluster ($K2^{K47-H(C\rightarrow G)}$) also abolished the HV phenotype of the strain carrying the intact cluster ($K2^{K47-H}$) (**Fig 7H**). To a less extent, removing the IS sequence from the K47-H *cps* cluster ($K2^{K47-H\Delta IS}$) also significantly reduced the virulence. The mortality data of these K2 variants were corroborated by their differences in surviving the clearance from the bloodstream (**Fig 7I**) and hepatic capture (**Fig 7J and 7K**) in the early phase of septic infection. Mice infected with $K2^{K47-H}$ showed a significantly higher level of bacteremia than those infected with $K2^{K47-L}$ and $K2^{K47-H}$ variants with the C-G mutation ($K2^{K47-H(C\rightarrow G)}$) or IS deletion ($K2^{K47-H\Delta IS}$). These findings revealed that both the IS motif and intact Wzc protein are required for the HV phenotype.

## Vaccination enables KCs to capture HV *K. pneumoniae* for a protective immunity

To assess if vaccination can facilitate the clearance of the HV *K. pneumoniae*, we immunized mice with inactivated whole cell vaccine (formaldehyde treated ATCC43816). The immunization significantly boosted the production of IgG antibodies against *K. pneumoniae* (**Fig 8A**). Upon challenge with three lethal doses of ATCC43816 ($10^4$, $10^5$ and $10^6$ CFU), all of naïve mice succumbed to the infection within 12 hr. By comparison, the immunized mice survived the challenge in a dose-dependent manner (**Fig 8B**). While the vaccine fully protected immunized mice from infection with $10^4$ CFU of ATCC43816, it only conferred 60% protection against the infection dose of $10^5$ CFU. Although none of the immunized mice survived the challenge against $10^6$ CFU, they showed significantly increase in the mean survival time (from 0.5 to 3.1 days). The immunoprotection was also reflected by reduced bacteremia in immunized mice (**Fig 8C**). The naïve mice rapidly developed severe bacteremia within 6 hr post i.p. infection with all the 3 doses as exemplified by high bacterial load levels of $3 \times 10^5$, $8 \times 10^6$ and $6 \times 10^7$ CFU/ml for infection doses of $10^4$, $10^5$ and $10^6$ CFU, respectively, at 6 hr. By contrast,

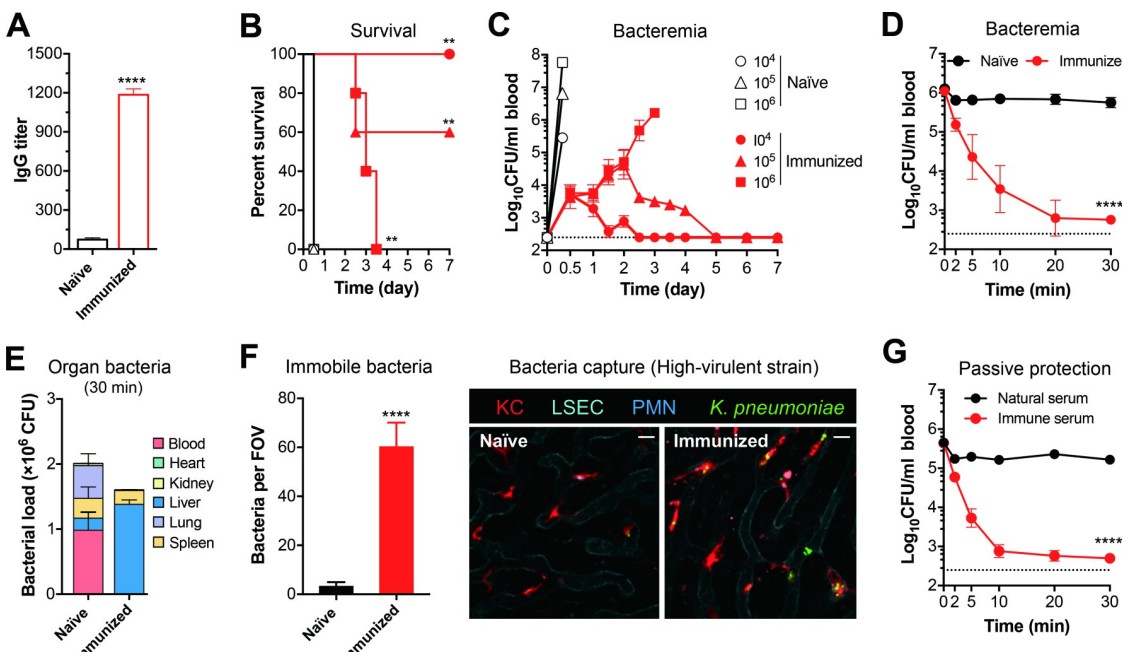

**Fig 8. Vaccination to boost capture of HV *K. pneumoniae* by KCs. A.** Anti-*K. pneumoniae* IgG titer in vaccinated mice. CD1 mice were immunized twice, one week apart, by subcutaneous injection of $10^8$ formaldehyde inactivated ATCC43816). Sera were collected from untreated (naïve) or immunized mice by retro-orbital bleeding two weeks after the second immunization and antibody titers against ATCC43816 were evaluated by ELISA. n = 3. **B, C.** Immunoprotection against HV *K. pneumoniae* infection. CD1 mice were immunized as in (A), and i.p. infected with different lethal doses ($10^4$, $10^5$, or $10^6$ CFU) of ATCC43816 to determine survival (**B**) and bacteremia levels (**C**). n = 5. **D, E.** Impact of immunization on the early clearance of HV *K. pneumoniae* by the liver. Bacteremia kinetics (**D**) and bacterial distribution (**E**) in the blood and five major organs of naïve or immunized mice at 30 min post i.v. infection with $10^6$ CFU of ATCC43816 are presented. n = 3. **F.** Vaccine-activated pathogen capture by Kupffer cells. Pathogen capture by KCs in the liver sinusoids of naïve and vaccinated mice was assessed by IVM post i.v. infection with $5 \times 10^7$ CFU of ATCC43816 as in Fig 5F. Six random fields of IVM images were used to quantify KC-associated bacteria per field of view (FOV) (left panel). Representative IVM images (right panel) are shown at the right panel. The process of bacterial capture is demonstrated in supplemental S4 Movie. Scale bar, 10 μm. n = 2. **G.** Immune serum-mediated clearance of HV *K. pneumoniae*. Naïve CD1 mice were i.v. infected with $10^6$ CFU of ATCC43816 pre-incubated with 25% natural or immune serum for 10 minutes at room temperature to determine bacteremia kinetics in the first 30 min. n = 5. Dash line denotes the detection limit. All data are presented as mean ± SD. Unpaired *t* test (A and F), log-rank test (B) and two-way ANOVA with Tukey's multiple comparisons test (D and G) were performed. **, $P < 0.01$; ****, $P < 0.0001$.

the immunized counterparts showed bacteremia levels below $10^4$ CFU/ml within 12 hr after the challenges. In particular, the immunized group kept bacteremia level below $10^5$ CFU/ml in the first two days post infection with $10^6$ CFU. This result demonstrated that immunization with inactivated *K. pneumoniae* confers significant protection against lethal infection of HV *K. pneumoniae*.

Vaccine-elicited immunity was also corroborated by effective clearance of HV *K. pneumoniae* in immunized mice at the beginning of septic infection. The i.v. inoculated ATCC43816 in immunized mice were rapidly cleared to 626 CFU/ml blood from the inoculum of $10^6$ CFU/ml at 30 min (**Fig 8D**). In sharp contrast, the bacteria were still abundantly present in the bloodstream of naïve mice ($10^6$ CFU/ml). Enumeration of bacteria in the major organs localized 86.0% of viable bacteria to the livers of immunized mice at 30 min, whereas only 8.9% of the bacteria were found in the livers of naïve mice at this time point (**Fig 8E**). These data demonstrated that vaccination empowers the livers to capture HV *K. pneumoniae* that are otherwise too "slippery" to be caught in naïve mice.

IVM imaging of the liver sinusoids uncovered that KCs are responsible for the vaccine-elicited capture of HV *K. pneumoniae* (**Fig 8F** and **S4 Movie**). Enumeration of immobilized

bacteria in the selected liver sinusoids identified 57 bacteria associated with KCs of immunized mice versus one bacterium on KCs of naïve mice (**Fig 8F**, left panel). Quantitative analysis of the IVM data showed that virtually all of the immobilized K2 bacteria in the liver sinusoids of vaccinated mice were associated with KCs, demonstrating that KCs are the major immune cells capturing the HV bacteria after immunization. Although splenic neutrophils of mice are shown to kill encapsulated *Streptococcus pneumoniae* [67], neutrophils were occasionally seen to pass the vasculatures with obvious association with circulating or immobilized bacteria. The K2 bacteria were occasionally deposited on the vascular wall of the liver sinusoids without apparent association with KCs (**S4 Movie**). This might be caused by spatial out-of-focusing effect of bacterium-associated KCs or sporadic binding of the bacteria to the vascular walls. We finally tested if vaccine-elicited antibodies are responsible for the KC capture of HV *K. pneumoniae* in immunized mice by pretreating K2 bacteria with immune serum from vaccinated mice before i.v. infection. CFU counts of the blood samples showed rapid clearance of immune serum-opsonized bacteria from the circulation, whereas natural serum-treated bacteria were well maintained in the bloodstream (**Fig 8G**). These results demonstrate that the vaccine enables the liver resident macrophages to capture the HV *K. pneumoniae* and thus protects the otherwise susceptible hosts from lethal outcome of blood-borne infection by HV *K. pneumoniae*.

## Discussion

### The liver—the major target organ of *K. pneumoniae* capsules

The liver has been recognized as an important organ for the clearance of pathogenic bacteria from the bloodstream [46–49, 68], but the specific impact of bacterial capsules on hepatic clearance has not been previously reported. This study shows that the anti-bacterial function of the liver can be bypassed by bacterial capsules in a capsule type- or structure-dependent manner. Significant advantage of encapsulated *K. pneumoniae* over isogenic acapsular mutant in the resistance to hepatic capture and killing unequivocally demonstrates the overall importance of the capsule in *K. pneumoniae* adaptation to host innate immunity. The benefit of the capsule toward *K. pneumoniae* survival in the blood circulation of mammalian host is further manifested by striking differences of the HV and LV capsule types in protecting *K. pneumoniae* from being captured by the liver. It is our view that the HV capsule types that are capable of fully bypassing the hepatic capture represent the rare "breakthrough" products of bacterial capsule evolution under the selection pressure of potent anti-bacterial activity in the liver. This opinion is supported by our observation that the liver is the most effective organ to capture and kill the vast majority of *K. pneumoniae* capsule types (e.g., the low-virulence capsules). A larger proportion of HV capsule types in the spleen than the liver indicates the spleen as the second line of defense against encapsulated HV bacteria, which agrees with the clinical observation that humans with splenic dysfunctions are particularly susceptible to encapsulated bacteria [55]. It remains to be determined whether the accumulation of HV *K. pneumoniae* in the spleen represents a result of active bacterial capture by splenic cells or the overspill of blood-borne bacteria in the absence of hepatic clearance.

### Kupffer cell—the primary target phagocyte of bacterial capsules in the liver

Capsules are known for the ability to inhibit host phagocytosis of bacterial pathogens. It is currently believed that the main function of capsules is to act on the circulating phagocytes during blood-borne infections of encapsulated bacteria [12]. By selective depletion of major phagocytes in mice (e.g., macrophage, neutrophil and monocyte) and visualization of bacterial interactions with host cells in the liver sinusoids, we have uncovered the liver resident macrophage

Kupffer cells as the primary target phagocyte of *K. pneumoniae* capsule, the major phagocyte for hepatic capture of the circulating microbes [46–49, 68, 69]. We have presented multiple lines of evidence that the low-virulence capsule types are partially successful in bypassing the recognition and capture of KCs as compared with the acapsular form; the high-virulence types represent the "elite" winners in the capture and anti-capture race between *K. pneumoniae* and KCs. This conclusion is consistent with the dominance of *K. pneumoniae* and other encapsulated bacteria in blood-borne infections [12].

## KC recognition of the LV capsule types–an anti-immune evasion mechanism

Capsule type-specific inhibition of free CPS against hepatic capture of the LV *K. pneumoniae* indicates that KCs recognize the LV capsule types via specific receptors. Although no KC receptor has yet been found to recognize bacterial capsules, significant blocking of poly(I) against KC recognition of K23 capsule strongly suggests that a novel scavenger receptor(s) on KC recognizes the capsule, which does not appear to involve the complement system since C3-deficient mice did not show significant impairment in hepatic capture of K23 *K. pneumoniae*. Complement protein C3 has been shown to accelerate the clearance of blood-borne *Escherichia coli* [70] and *Streptococcus pneumoniae* [41] in the liver and spleen of rodents. Jensen et al. have recently reported the deposit of complement C3 on the capsule of certain *K. pneumoniae*, which causes morphological change of the capsule structure [71]. In this context, we cannot exclude the possibility that the complement system is necessary for KC capture of other *K. pneumoniae* capsule types. Numerous pattern recognition receptors (PRR) have been shown to recognize pathogen-associated molecular patterns (PAMP) on the repeating units of bacterial capsular polysaccharides [12]. Alveolar macrophages possess a receptor(s) that specifically bind to the Manα2/3Man or Rhaα2/3Rha disaccharide motif on multiple types of *K. pneumoniae* [72]. The surfactant protein A (SP-A) binds to K21 but not K2 capsule by a specific recognition of a mannose configuration on K21 capsule [73]. Human L-ficolin recognizes acetylated monosaccharides on the capsules of many bacteria [74–76]. Capsule binding of soluble lectins (e.g., SP-A and ficolin) has been shown to activate the complement system by the lectin pathway and thereby enhance opsonophagocytosis of encapsulated bacteria [77–80]. However, the complement system does not appear to be involved in the recognition of K23 capsules based on the complete dispensability of C3 and CRIg in hepatic capture of K23 *K. pneumoniae*. In the context of host recognition and pathogen anti-recognition race, recognition of the LV capsule types by KCs appears to represent a less effective but still functional strategy of the macrophages after the more "vulnerable" PAMPs are covered by the capsule (e.g., LPS, cell wall peptidoglycan, and lipoproteins). This statement is supported by our observation that *K. pneumoniae* acapsular mutant is recognizable by multiple host factors (e.g., CRIg and scavenger receptor), but K23 is only recognized by a scavenger receptor(s).

## Causal relationship between *K. pneumoniae* capsule type and virulence

Previous clinical and epidemiology studies have shown remarkable diversity in genome sequence, virulence, capsule composition, antibiotic susceptibility and mucoid phenotype of *K. pneumoniae* [1, 2, 5]. Great variation in virulence among *K. pneumoniae* strains are also recapitulated in mouse models [5, 33, 81]. However, the molecular mechanisms governing the impact of genomic variation on *K. pneumoniae* virulence remain undefined. In this study, we demonstrate that variation in capsule type/composition directly impacts virulence phenotypes of this pathogen by using isogenic bacteria derivatives of a capsule type K2 strain, which avoided potential complications from variable genetic backgrounds of wild type strains.

Consistent with the dominance of K1 and K2 isolates in invasive infections [28–31], the isogenic K2$^{K1}$ and K2$^{K2}$ derivatives showed a high-virulence phenotype while infection with isogenic K2$^{K3}$ and K2$^{K23}$ strains yielded a low-virulence phenotype. Based on the data associated with the virulence and hepatic capture of clinical isolates (Figs 1A and S2D), K16 and K20 also belong to the HV capsule types, which agrees with the prevalence of K16 in hvKp isolates [12, 27]. The K2 capsule-enhanced resistance of the K3 (K3$^{K2}$) and K7 (K7$^{K2}$) strains to hepatic capture also support the importance of capsule type in the immune evasion of *K. pneumoniae*. Finally, the phenotypic differences of isogenic K47 HV (K2$^{K47-H}$) and LV (K2$^{K47-L}$) also argue for the significance of the capsule structure in shaping the fate and virulence of *K. pneumoniae*. While it requires further investigation to determine the precise molecular difference in CPSs of the HV and LV K47 strains, phenotypic impact of the two variations in the *cps* gene clusters on the hepatic capture and virulence traits suggests that the LV 47 capsule appears to be a "degenerate" form of the same HV capsule. In particular, the C-terminal Wzc truncation may alter the length of the K47 CPS or the number of capsule repeating unit, since the truncated C-terminal amino acids are implicated in Wzc autophosphorylation and thereby the polymerization and translocation of the Wzy-dependent CPSs [16, 66].

The poor capture of K1 and K2 *K. pneumoniae* by KCs in the liver is somewhat paradoxical to the prevalence of these capsule types in liver abscess [82]. A possible explanation is that the HV capsules confer *K. pneumoniae* with superior intracellular adaption once the bacteria enter KCs or other cells in the liver. This notion is supported by the previous report that K1 and K2 strains are more capable of surviving and replicating inside mouse and pig KCs than K17 and K107 strains, resulting in inflammatory responses in the liver [51]. The capsule has also been shown to enhance intracellular survival of *S. pneumoniae* [26]. A recent study has also documented that KCs confer a protective role against *K. pneumoniae*-induced liver abscess [50]. In this context, the precise role of KCs in the *K. pneumoniae*-induced liver disease warrants future investigation.

## The importance of non-*cps* genes in the immune evasion and virulence of *K. pneumoniae*

While this study has uncovered the profound impact of capsule types/structures on *K. pneumoniae* evasion of KC capture and virulence, our data also underscore the significance of *K. pneumoniae* strain background or non-*cps* genes on host clearance and bacterial virulence. This is manifested by differential contribution of the K2 *cps* gene cluster to hepatic clearance and virulence of the K2, K3 and K7 strains. While the K2 capsule transformed the acapsular K2 strain into a bacterium (K2$^{K2}$) with the elite survival capacity in the bloodstream and extreme virulence, the same capsule made significant but less potent impact on the acapsular K3 and K7 strains (e.g., K3$^{K2}$ and K7$^{K2}$). These phenotypic differences in virulence traits also correlate with the hypermucoviscosity status of these K2 capsule-producing variants. Introducing the K2 *cps* genes in the acapsular K2 mutant (hypermucoviscosity negative) made the recipient (K2$^{K2}$) produce colonies with hypermucoviscosity, but the same gene cluster failed to confer the acapsular K3 or K7 the same colony phenotype (e.g., K3$^{K2}$ and K7$^{K2}$). This finding echoes the previous observation that the *K. pneumoniae* capsule is necessary but not sufficient for the hypermucoviscosity phenotype [10, 15, 83]. Along this line, the K1 *cps* genes could not covert the acapsular K2 into a hypermucoviscosity strain (K2$^{K1}$) although the strain possessed the high virulence traits. The latter agrees with our observation that hypermucoviscosity is only associated with the majority of the HV *K. pneumoniae* isolates (**S1 Table**). This information may be applied to deciphering the molecular basis and pathogenic role of hypermucoviscosity in *K. pneumoniae*.

### Vaccine against HV *Klebsiella*

Despite the current lack of a licensed vaccine for *K. pneumoniae*, many attempts have been made in vaccine development with limited information in active protection in animal models [34, 35]. A bio-conjugate vaccine consisting of K1 and K2 CPS protects mice from lethal respiratory infection of highly virulent *K. pneumoniae* [34]. To the best of our knowledge, the inactivated *K. pneumoniae* cells presented here is the first case of whole cell vaccine that confers protection against lethal bloodstream infection of highly virulent *K. pneumoniae*. Since the immunization made circulating HV *K. pneumoniae* adhere to KCs, it is highly likely that vaccine-elicited antibodies mediated the binding interactions by their opsonizing effect. Antibodies against the *K. pneumoniae* capsules have been shown to enhance killing of *K. pneumoniae* under the *in vit*ro and *in vivo* conditions [36–38]. It is thus likely that some of the KC-enabling effect was derived from capsule-specific antibodies. This notion is consistent with the fact that immunization with *K. pneumoniae* capsule variants is able to induce protective antibodies [34]. Although the precise mechanism for the protective immunity of this *K. pneumoniae* vaccine requires further investigation, our finding provides a proof of concept for future development of vaccines for HV *K. pneumoniae*.

## Materials and methods

### Ethics statement

All mouse experiments were approved by the Institutional Animal Care and Use Committee in Tsinghua University under the protocol 14ZJR-1.

### Bacterial strains and cultivation

All the *K. pneumoniae* clinical isolates used in this study were originally obtained from the blood or other sterile specimens (ascite, bile, bone marrow, CSF, joint fluid, pus, muscular tissue and midstream urine) of human patients in four Chinese hospitals as described in **S1 Table**. The laboratory strains and their derivatives from this work are listed in **S3 Table**. *Escherichia coli* DH5α was used for DNA cloning as described [84]. *K. pneumoniae* and *E. coli* strains were grown in Luria-Bertani (LB) broth or on LB agar plates as described [85]. When necessary, the following antibiotics or chemicals were added: apramycin (30 μg/ml), spectinomycin (300 μg/ml), arabinose (0.2%) and sucrose (5%).

### Chemicals and reagents

All ingredients of bacterial culture media and chemicals were obtained from Sigma (Shanghai, China) unless otherwise stated; all the molecular biology reagents obtained from New England Biolabs (Beijing, China). The primers used in this study were synthesized by Ruibio Biotech (Beijing, China) and are described in **S4 Table**.

### Capsule typing

Capsule types were genotypically determined by a multiplex PCR-based approach as described [86]. The *wzi* gene was amplified and sequenced with primers Pr14753 and Pr14754. The resulting sequences were used to search for the *wzi* alleles and corresponding capsule types (K or KL) at https://bigsdb.pasteur.fr/klebsiella as described [87].

### Mutagenesis

Capsule type switching was performed by replacing the *cps* gene cluster of the recipients with the amplicons of target capsule types using a CRISPR-based mutagenesis approach as

described [85] and elaborated in **S1 Fig** and **S5 Table**. Capsule switching was carried out in strains ATCC43816 (K2), TH12849 (K3) and TH12880 (K7) by replacing the entire endogenous *cps* gene clusters with the corresponding sequences of the donor strains using the flanking regions as the basis for homologous recombination. The resulting hybrid strains also contained the approximately 400-bp upstream and 200-bp downstream sequences of the donor *cps* genes, which were predicted to include the major functional DNA elements of the donor *cps* genes (e.g., promoter and regulatory sequences). ATCC43816 is highly virulent in mice with an $LD_{50}$ of less than 100 CFU as previously determined by intratracheal infection [52]. TH12849 and TH12880 were identified as the LV strains as listed in **S1 Table**. Due to the complex nature of the genetic modifications associated with the recombinations between the donor and recipient *cps* loci, we utilized three different strategies to make the *cps* gene replacements. The details of the mutagenesis in the *cps* loci and resulting strains are described in **S1 Fig** and **S5 Table**. The transformants were selected with apramycin and spectinomycin at 30˚C; the plasmids cured with 5% sucrose at 37˚C. The sequences of the *cps* locus in all the engineered strains were verified by DNA sequencing. The plasmid used in this study are described in **S6 Table**.

## Bacterial genome sequencing and DNA analysis

The complete genomes of *K. pneumoniae* isolates TH12908 (K1), TH12852 (K23), TH12880 (K7), TH12845 (K47) and TH12846 (K47) were sequenced by the PacBio Sequel platform as described [88], and are available in NCBI under bioproject PRJNA778913 and PRJNA846980. All in-house DNA sequence analyses were carried out by the DNASTAR Lasergene v10 for Macintosh (Madison, WI).

## Quantification of capsule production and mucoviscosity

The capsule production was quantified by measuring uronic acid (UA) content as described previously [54]. The bacteria were cultured in LB medium overnight, transferred in fresh medium at 37˚C for 6 hr incubation and measured the optical density at 600 nm ($OD_{600}$). Capsule was extracted from 500 μl bacterial culture with Zwittergent 3–14 detergent (Sigma-Aldrich), precipitated with ethanol and dissolved in tetraborate/sulfuric acid. After the incubation with 3-hydroxydiphenol, the absorbance of the mixture at 520 nm was measured. The UA content was quantified from a standard curve of glucuronic acid and presented as micrograms per $OD_{600}$.

Hypermucoviscosity of *K. pneumoniae* colonies was assessed by the string test as previously described [5]. Strains that produced colonies on blood agar plates with a viscous string of ≥ 5mm were regarded as hypermucoviscosity positive. The sedimentation assay was used as described previously to quantify mucoviscosity [89]. The bacterial cultures used in the UA assay were centrifuged for 5 min at 1,000 g and the $OD_{600}$ of the supernatant was determined by spectrophotometry. The mucoviscosity status was assessed by the ratio of supernatant $OD_{600}$ to bacterial culture $OD_{600}$ before centrifugation.

## Immune cell depletion

Depletion of neutrophils and neutrophils/inflammatory monocytes were accomplished by i.p. injection of 500 μg anti-Ly6G antibody (1A8, BioXCell) and 500 μg anti-Ly6G/Ly6C antibody (Gr1, BioXCell), respectively, 24 h before infection as described [90]. Macrophages were depleted by i.p. inoculation with 1 mg clodronate liposomes (CLL) 72 h prior to infection as described [91]. Depletion of KCs in *Clec4f*-DTR mice was achieved by i.v. injection of 25 ng/g diphtheria toxin 24 h before infection as described [57].

## Flow cytometry

Immune cell depletion was assessed by flow cytometry as described [57]. First, the cell suspensions of blood, spleen and liver were prepared as follows. Blood immune cells were collected by lysis of 200 μl of whole blood with 5 ml RBC lysis solution (BioLegend) and centrifugation at 300 g for 5 min. Splenocytes were isolated as previously reported [67]. Briefly, the spleen was passed through the 70-μm strainer and resuspended in 5 ml FACS buffer (PBS with 3% FBS). The residual red blood cells were lysed by 1 ml RBC lysis solution. Liver non-parenchymal cells (NPCs) were prepared by as described [92]. Briefly, the liver of euthanized mice was perfused with 5 ml of digestion buffer (HBSS with 0.5 mg/ml collagenase IV, 0.5 mM $CaCl_2$ and 20 μg/ml DNase I) from the portal vein. The perfused liver was shredded, suspended in 10 ml digestion buffer, and then rotated continuously at 300 rpm for 30 minutes at 37°C. The liver homogenate was passed through the 70-μm strainer and chilled on ice for 5 min, washed by centrifugation and resuspended in cold HBSS. The cells in the suspension were collected by centrifugation at 300 g for 5 min. The residual red blood cells were lysed by 1 ml RBC lysis solution and the hepatocytes were removed by centrifugation at 50 g for 2 min. The liver NPCs in the supernatant were obtained by centrifugation at 300 g for 5 min at 4°C. The Fc receptors of immune cells from 200 μl blood, splenocytes ($10^6$) and liver NPCs ($10^6$) were blocked in 50 μl FACS buffer with 1% anti-CD16/32 antibody for 10 min and stained with antibodies: 0.25 μl anti-F4/80 (FITC), 0.25 μl anti-CD45 (APC-Cy7), 0.1 μl anti-CD11b (BV605), 0.25 μl anti-CD31 (APC), 0.1 μl anti-Ly6G (PE) and 0.1 μl anti-Ly6C (PB). Cells were finally stained with 5 μl 7-AAD for the viability before analysis. The populations of viable neutrophils ($CD45^+CD11b^+Ly6C^+Ly6G^{high}$), inflammatory monocytes ($CD45^+ CD11b^+Ly6C^{high}Ly6G^-$), splenic red pulp macrophages (RPMs) ($CD45^+CD11b^{low}F4/80^{high}$) and KCs ($CD45^+CD31^-CD11b^{low}F4/80^{high}$ were gated.

## Intravital microscopy (IVM)

IVM imaging of mouse liver was accomplished with inverted confocal laser scanning microscope as described [91]. Briefly, the liver sinusoid endothelial cells, KCs and neutrophils were labelled by i.v. injection of 2.5 μg AF594 anti-CD31 (MEC13.3, BioLegend, USA), AF647 anti-F4/80 (BM8, BioLegend, USA) and 2.5 μg PE anti-Ly6G (1A8, eBioscience, USA) antibodies, respectively, 30 min before i.v. infection with $5 \times 10^7$ CFU of FITC-labeled *K. pneumoniae*. FITC labeling was performed by incubating bacteria in 100 μl phosphate buffer saline (PBS) with 20 μg FITC (Sigma) for 30 min at room temperature in dark and washing three times with PBS as described [93]. Double labeling with pHrodo and FITC was performed by incubating bacteria in 100 mM $NaHCO_3$ (pH 8.5) buffer with 0.001 mM pHrodo Red dye (Thermo Fisher Scientific) and 200 μg/ml FITC for 30 min at 4°C in the dark, according to the manufacturer's instructions. The double-labeled bacteria were washed three times with the $NaHCO_3$ buffer to remove free dyes before infection. Leica TCS SP8 confocal microscope with a 10×/ 0.45 NA and 20×/0.80 NA HC PL APO objectives was used to acquire images. The fluorescence signals were detected using Photomultiplier tubes (PMTs) and Hybrid Photo detectors (HyD) (600 × 600 pixel size for time lapse series and 1,024 × 1,024 pixel size for representative photographs). Four laser excitation wavelength channels (488, 535, 594 and 647 nm) were applied by white light laser (1.5 mw, Laser kit 1 WLL2, 470–670 nm). Real time imaging was monitored for 3 to 5 min after infection. Multiple images were collected consecutively in the same fields of view with an interval of 2 seconds; bacteria without positional change in multiple images were defined as immobilized bacteria. Five to six random fields of view were collected to calculate the bacterial number per field of view (FOV).

## CPS purification

CPS were extracted from LB broth cultures of *K. pneumoniae* and quantified as described by Kobayashi *et al.*[36].

## Mouse infection

Septic infections were performed in female CD1 or C57BL/6 mice (6–8 weeks old) by i.v. or i.p. inoculation of *K. pneumoniae* as described [94]. Briefly, the bacteria from pre-counted frozen stocks were collected by centrifugation and resuspended in Ringer's solution to the desired concentration in the final volume of 200 μl for i.p. and 100 μl for i.v. inoculations. CFU counts of each inoculum was enumerated by plating its various dilution before infection to retroactively calculate the actual infection dose. The infection dose and route of each experiment are specified in the figure legends. Splenectomy and sham operations in mice were carried out as described [95], and used for infection experiments 7 days post the surgery. Briefly, before surgery, the mice were anesthetized with 400 mg/kg of avertin (Sigma-Aldrich) by i.p. and subcutaneous injection with 8 mg/kg of meloxicam (Sigma-Aldrich). The incision on the left side of peritoneum was generated to ligate the splenic pedicle and remove the spleen before closing the peritoneal incision. The sham-operated (SHM) mice were treated in the same way without splenic removal. The *Clec4f*-DTR mice were generated as described [57]. $C3^{-/-}$ and $CRIg^{-/-}$ [46] mice were acquired from the Jackson Laboratory and Genentech, respectively. The survival rate was determined in 7 days post infection. Bacteremia level was characterized by CFU plating of blood samples obtained by retro-orbital bleeding. Bacterial level in the blood at 0 min was estimated by dividing CFU counts of the inoculum with the total blood volume of mice, which as calculated according to 0.07 ml of blood per gram weight of mouse [96]. Bacteria in organs were similarly determined by plating tissue homogenates and extracting CFU of residual blood of individual organs. The volume of residual blood was estimated as 0.1 ml of blood per gram of a well-perfused organ [97]. The total bacteria per mouse was calculated as the sum of CFU values derived from the blood and organ samples. The percentage of inoculum was calculated by dividing the total number of viable bacteria by the inoculum. Impact of free CPS, polyinosinic acid (poly(I)) or polycytidylic acid (poly(C)) on bacterial clearance was tested by i.v. inoculation of these materials 2 min before infection.

## Immunization

Immunoprotection of formaldehyde inactivated whole cell vaccine against HV *K. pneumoniae* was assessed in female CD1 mice (6–8 weeks old) essentially as described [98]. Briefly, freshly culture of ATCC43816 ($10^8$ CFU) was collected by centrifugation and washed twice with PBS. The pellet was suspended in 0.5% formaldehyde at $10^9$ CFU/mL and incubated at 4˚C overnight. Inactivated bacteria were washed twice with PBS by centrifugation and resuspension, resuspended to $2 \times 10^8$ CFU in 500 μl PBS containing 25 μl 0.5% aluminum phosphate adjuvant, and used for immunization by subcutaneous injection of $10^8$ CFU inactivated bacteria. The same immunization was repeated once 7 days later. The protection efficacy was determined by i.p. infection with lethal doses of ATCC43816 14 days after the second immunization; mice periodically sampled for blood CFU by retroorbital bleeding, and daily monitored for survival for 7 days. Passive protection of immune serum against HV *K. pneumoniae* was assessed as described [99]. ATCC43816 bacteria ($10^6$ CFU) were incubated with 25 μl heat-inactivated serum collected from naïve (natural) mice or mice immunized with the whole cell vaccine (immune) in total volume of 100 μl PBS for 10 minutes at room temperature before i.v. inoculation into mice and enumeration of bacteria in the peripheral blood.

## Antibody detection

IgG titers of immune sera were determined by enzyme-linked immunosorbent assay (ELISA) as described [85]. Briefly, ATCC43816 was grown to $OD_{600}$ 0.1 before being pelleted and resuspended in the same volume of PBS. Each well of 96-well plates was coated with 100 μl bacterial suspension, and incubated overnight at 4˚C, and blocked with 200 μl 5% non-fat dry milk (BD, USA). Mouse serum was serially diluted in PBS and added to appropriate wells incubating at 37˚C for 2 h. Each well was washed three times with PBS, and bound antibody was detected with alkaline phosphatase-conjugated goat anti-mouse IgG (1:2,000, EasyBio, Beijing, China) at 37˚C for 1h before addition of the TMB substrate (TIANGEN, Beijing, China). The optical absorbance was measured at a 450 nm wavelength using the BioTek Synergy H1 microplate reader (BioTek, Germany).

## Statistical analysis

All statistical analyses were carried out with the GraphPad Prism software (version 7.0a for Mac) and all data are shown as mean ± SD. The specific analysis for each figure is noted in the figure legends.

## Supporting information

**S1 Fig. Schematic diagram of experimental procedure for genetic replacement of cps locus in *K. pneumoniae*.** (**A**) Construction of capsule isogenic variants K2^K1, K2^K2, K2^K3 and K2^K23. The *cps* cluster of ATCC43816 was divided into three segments indicated as the pink, red and crimson regions to delete sequentially in the same manner (step 1–6). (1) Deletion of the first segment of *cps* cluster. The spacer targeting *cps* cluster was cloned into sgRNA-expressed plasmid pSGKP-spe. The repair template for repairing Cas9-cleaved DNA by homologous recombination was constructed by fusion PCR with up- and downstream sequence of target region. Spacer-introduced plasmid and repair template were co-electroporated into Cas9-expressed strain generating strain with deletion of *cps* genes (Δ*galF-wzc*). (2) Curing the spacer-introduced plasmid. (3–7) Deleting the second segment (3 and 4) and third segment (5–7) of *cps* cluster generated Δ*cps* strain by the same procedure as in (1) and (2). (8–9) The *cps* cluster of other capsule types was transformed into the Δ*cps* strain to yield capsule replacement strain. The spacer-introduced plasmid expressing sgRNA to target junction of *cps* up- and downstream region and donor *cps* amplicons from 12908 (K1), ATCC43816 (K2), TH12849 (K3) or TH12852 (K23) strain were co-transformed to Δ*cps* strain generating capsule-switched strain. (**B**) Construction of capsule isogenic variants (K3^K3, K3^K2, K7^K7 and K7^K2) using LV recipient strains. (1–2) Deletion of the *cps* cluster. (3–4) Introduction of the 5' and 3' K2 homologous arms (1 kb each) and kanamycin resistance gene (*kan*^R) as screening marker to construct the transitional strain. (5–6) The K2 *cps* cluster of ATCC43816 was transformed into the transitional strain to yield capsule replacement strain K3^K2. The K7^K7 were constructed in the same way. (**C**) Construction of capsule isogenic variants K2^K47-L and K2^K47-H. (1–2) Construction of the transitional strain with K47 homologous arms and *kan*^R gene. (3–4) The *cps* cluster amplicons of TH 12845 (K47, high virulence) and TH 12846 (K47, low virulence) capsule types was transformed into the transitional strain to yield capsule replacement strain K2^K47-L and K2^K47-H.
(TIF)

**S2 Fig. Capsule type-dependent interception of circulating *K. pneumoniae* in the liver. (related to Fig 3).** Survival rate (**A**) and bacteremia kinetics in 6 hr (**B**) of the CD1 mice i.v. infected with $5 \times 10^6$ CFU of capsule isogenic variants derivatives of ATCC43816. n = 6.

Percentage of total viable bacteria at 5 min post infection to the initial inoculation (**C**). The total bacteria burdens of blood and five major organs divided by the initial inoculum are presented as percentage of inoculum. n = 6. Viable bacteria of wildtype *K. pneumoniae* strains representing 21 capsule types in organs and blood at 30 min (**D**). n = 1. Changes of bacterial load in blood (**D**), liver (**E**) and spleen (**F**) from 5 to 30 min post infection. n = 6. Viable bacteria of Δ*cps* (**G**), $K2^{K2}$ (**H**) or $K2^{K23}$ (**I**) in blood, liver and spleen within 6 hr after infection. n = 6. Bacterial load in blood and organs (**K**) and survival rate (**L**) of mice infected with capsule-switched derivatives of LV strain at 30 min. n = 4–6. Capsule production (**M**) and mucoviscosity (**N**) of capsule hybrid strains ($K3^{K2}$ and $K7^{K2}$), the related LV recipients (K3 and K7) and HV donor (K2). The data are presented as mean ± SD. One-way ANOVA with Tukey's (**C**) or Dunnett's (**M** and **N**) multiple comparisons test, and two-way ANOVA with Sidak's (**E-G**) multiple comparisons test were performed. ns, not significant; *, $P < 0.1$; **, $P < 0.01$; ***, $P < 0.01$; ****, $P < 0.0001$.
(TIF)

**S3 Fig. Flow cytometry analysis of specific cell depletion.** Depletion efficiency of neutrophil and monocyte (**A**). Mice were treated with ISO (isotype control), 1A8 antibody (depleting neutrophil) or Gr1 antibody (depleting neutrophil and inflammatory monocyte). The proportion of neutrophils ($Ly6C^{low}/SSC^{high}$) and inflammatory monocytes ($Ly6C^{high}/SSC^{low}$) in blood myeloid population ($CD45^{+} / CD11b^{+}$) was measured. CD1, n = 2. Depletion efficiency of tissue resident macrophage in spleen and liver (**B, C**). The ratios of RPM and KC ($CD11b^{low}/F4/80^{+}$) in the immune cells ($CD45^{+}$) of CD1 mice treated with PBS or CLL (depleting macrophages) for 72 hr (**B**) and *Clec4f*-DTR mice treated with or without DT (specific depleting KC) for 12 hr (**C**) were measured. n = 2. The data are presented as mean ± SD.
(TIF)

**S4 Fig. Inhibition of *K. pneumoniae* capture in the liver by free CPS.** 50% clearance time of LV *K. pneumoniae* (K23) in mice with CPS pretreatment (related to Fig 5) (**A**). Time to clearance of 50% inoculum from blood was calculated based on nonlinear regression analysis of early bacteremia data. CD1, n = 3. Effect of free K2 capsule on clearance of HV K2 strain (**B**). Mice were intravenously inoculated with PBS or 800 μg purified CPS from the K2 strain 2 min before i.v. infection with $10^5$ CFU of K2. The bacterial loads in blood within 30 min were presented. CD1, n = 5. The data are presented as mean ± SD. Ordinary one-way ANOVA with Tukey's multiple comparisons test (**A**) and two-way ANOVA with Tukey's multiple comparisons test were performed (**B**). *, $P < 0.1$; **, $P < 0.01$, ****, $P < 0.0001$.
(TIF)

**S1 Table. Virulence characteristics of *K. pneumoniae* isolates.**
(XLSX)

**S2 Table. Bacterial load in blood and organs.**
(XLSX)

**S3 Table. Laboratory strains and derivatives used in this study.**
(DOCX)

**S4 Table. Primers used in this study.**
(DOCX)

**S5 Table. Information for constructions of strains used in this study.**
(DOCX)

**S6 Table. Plasmids used in this study.**
(DOCX)

**S1 Movie. IVM shows KC capture of acapsule (Δ*cps*) and LV (K2<sup>K23</sup>) variants but not HV (K2<sup>K2</sup>) *K. pneumoniae*.** The KC (red), LSEC (cyan), and *K. pneumoniae* (*Kpn*, green) in the liver approximately 10 min post i.v. infection with $5 \times 10^7$ CFU of ATCC43816 derivatives were shown. Graphic analysis was shown in Fig 5E.
(MOV)

**S2 Movie. IVM detection of acapsule *K. pneumoniae* capture by KCs of *Clec4f*-DTR mice treated with or without DT.** Graphic analysis was shown in Fig 5F.
(MOV)

**S3 Movie. IVM detection of LV (K2<sup>K23</sup>) *K. pneumoniae* capture by KCs of *Clec4f*-DTR mice treated with or without DT.** Graphic analysis was shown in Fig 5F.
(MOV)

**S4 Movie. IVM detection of HV (K2) *K. pneumoniae* capture by KCs of vaccinated mice.** Graphic analysis was shown in Fig 8F.
(MOV)

## Acknowledgments

We thank Dr. Bei Li at Hubei University of Medicine for sharing strain NTUH-2044, Tsinghua research platforms for assistance in animal experimentation (Laboratory Animal Research Center), and flow cytometry and IVM imaging (Center for Cell Biology).

## Author Contributions

**Conceptualization:** Jing-Ren Zhang.

**Data curation:** Xueting Huang, Xiuyuan Li.

**Formal analysis:** Xueting Huang.

**Funding acquisition:** Xueting Huang, Haoran An, Yuan He, Jing-Ren Zhang.

**Investigation:** Xueting Huang, Xiuyuan Li, Juanjuan Wang, Ming Ding, Lulu Li.

**Methodology:** Xueting Huang, Xiuyuan Li, Haoran An, Juanjuan Wang, Jing-Ren Zhang.

**Project administration:** Xueting Huang, Jing-Ren Zhang.

**Resources:** Lijun Wang, Quanjiang Ji, Fen Qu, Hui Wang, Yingchun Xu, Xinxin Lu.

**Supervision:** Haoran An, Jing-Ren Zhang.

**Validation:** Jing-Ren Zhang.

**Writing – original draft:** Xueting Huang, Xiuyuan Li.

**Writing – review & editing:** Xueting Huang, Haoran An, Jing-Ren Zhang.

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
