## [Decision Letter · Decision Letter 0]

24 Jan 2022

Dear Dr. Zhang,

Thank you very much for submitting your manuscript "Capsule-Mediated Evasion of Kupffer Cell Capture Defines Klebsiella pneumoniae Virulence" for consideration at PLOS Pathogens. As with all papers reviewed by the journal, your manuscript was reviewed by members of the editorial board and by several independent reviewers. In light of the reviews (below this email), we would like to invite the resubmission of a significantly-revised version that takes into account the reviewers' comments.

The reviewers and editors appreciate the potentially important findings in these studies. However, all reviewers and editors note that more experimental details are needed, such as characterization of the mucoviscosity of the strains used in the study to further delineate whether the phenotypes observed are due to changes in amount of capsule produced. In addition, significant concerns were raised that only a few mice were used for the experiments (N=3) and that the experiments may have only been done once. The experiments should be repeated twice and combined so more mice are evaluated. Other concerns are raised by the reviewers including including CFU per organ rather than just percentage, clearer description of what is significant and what comparisons are being made. If you chose to resubmit to PloS Pathogens, please address all the concerns of each of the reviewers in your revised manuscript.

We cannot make any decision about publication until we have seen the revised manuscript and your response to the reviewers' comments. Your revised manuscript is also likely to be sent to reviewers for further evaluation.

Sincerely,

Joan Mecsas

Associate Editor

PLOS Pathogens

Christoph Tang

Section Editor

PLOS Pathogens

Kasturi Haldar

Editor-in-Chief

PLOS Pathogens

orcid.org/0000-0001-5065-158X

Michael Malim

Editor-in-Chief

PLOS Pathogens

orcid.org/0000-0002-7699-2064

The reviewers and editors appreciate the potentially important findings in these studies. However, all reviewers and editors note that more experimental details are needed, such as characterization of the mucoviscosity of the strains used in the study to further delineate whether the phenotypes observed are due to changes in amount of capsule produced. In addition, significant concerns were raised that only a few mice were used for the experiments (N=3) and that the experiments may have only been done once. The experiments should be repeated twice and combined so more mice are evaluated. Other concerns are raised by the reviewers including including CFU per organ rather than just percentage, clearer description of what is significant and what comparisons are being made. If you chose to resubmit to PloS Pathogens, please address all the concerns of each of the reviewers in your revised manuscript.

Reviewer's Responses to Questions

**Part I - Summary**

Reviewer #1: The manuscript “Capsule-Mediated Evasion of Kupffer Cell Capture Defines Klebsiella pneumoniae Virulence” by Huang, et al. presents data suggesting that Kupffer cells of the liver are a primary source for eliminating low virulence K. pneumoniae (Kp), and that the ability of Kupffer cells to respond to Kp is dependent on capsule type. This is a potentially interesting study, pointing to the liver as a primary organ of defense against Kp. They further perform experiments examining the recognition of capsule by Kupffer cells, and at the potential for use of inactivated bacteria as a vaccine against Kp infection. For the bulk of this work, the experiments seem to appropriate for the questions posed and the paper is reasonably well-written given the presumed challenge of writing in a second language. Swapping the capsule loci, replacing a HV K-type with a LV K-type, is a creative approach to examine the role of capsule type in the severity of disease. There are, however, some critical control experiments that are needed for accurate interpretation of the data generated from these hybrid strains.

A major weakness of this manuscript is the near complete omission of the contribution of mucoviscosity to hypervirulent Kp. The parental strain (43816) for the hybrid K type strains is known to be hypermucoviscous (HMV). Because it is not known if HMV is dependent on capsule type, it is essential that capsule levels and mucoviscosity be quantified in the strains used. If the hybrid strains produce little capsule (less than donor strain) or are HMV negative, then the interpretation of capsule-type dependency is impossible to separate from capsule amount or HMV status. One potential strain to consider adding to the analysis is a hybrid strain with the K47 cps locus. As at least some of the K47 isolates were HV, this demonstrates that this capsule type has the potential for hypervirulence. The data presented within is interesting independent of the capsule levels/HMV status, but this information is critical for the current interpretation of capsule-type dependence for Kupffer cell clearance.

Reviewer #2: The paper by Huang et al examines the role of K. pneumoniae capsule variability and liver-resident macrophages (KCs) in determining the immune clearance of the pathogen. Certain capsule-types are associated with hypervirulence (HV) in clinical strains, while others show low virulence (LV). HV-associated capsules allow high early burdens in the bloodstream, while LV capsules lead to clearance from the blood and high early liver loads. Transgenic mice with KC-selective depletion revealed that KCs are a key determinant of the difference in clearance. Pre-injection with pure capsule delays liver capture of LV bacteria with the same capsule, consistent with capsule as a key surface component mediating capture. Immunization of mice with HV bacteria blocks development of HV and reverses the evasion of capture/clearance by liver KC cells.

Major strengths of the study include the combined analysis of capsule-variant clinical isolates as well as recombinant isogenic capsule-switched strains; and the elegant and systematic in vivo analyses taking advantage of the transgenic KC-depletion model and intravital microscopy. The paper expands our knowledge of the early events determining pathogenesis with invasive K. pneumoniae infections and how capsule variability modulates this process. There are a few areas for improvement, namely improved validation the capsule-switched strains as well as enhancing the text by discussing potential role of complement and relevant past studies.

Reviewer #3: This study has examined the role of capsule type in Klebsiella pneumoniae virulence. The authors found that certain capsule types are associated with increased virulence in a murine model infection. Some of these capsule types are also associated with hvKp human infections. To eliminate confounding issues of genetic background, the authors swapped out the K2 cps locus of ATCC 43816 with K1, K3, K23, or no locus (∆cps). ATCC 43816 with K1 or K2 still exhibited a high virulence (HV) phenotype and ATCC 43816 with K3 and K23 retained a low virulence (LV) phenotype. Using these same 5 strains, the authors show that HV strains are primarily found in the blood, while LV strains are retained in the livers and that HV strains are more resistant to killing in the blood than LV strains, which are more robust than ∆cps. Mice treated with clodronate liposomes have reduced ability to restrict K. pneumoniae infection to the livers, implicating macrophages with this phenotype. Targeted deletion of Kupffer cells (KCs) confirms that it is liver-resident macrophages responsible for restricting K. pneumoniae infection to the liver and that they co-localize with a LV K23 strain more than with an HV K2 strain. Treating mice with K23 CPS 2 min prior to infection reduces capture of K. pneumoniae in the liver. Vaccination with a killed-HV K2 strain protects mice from bacteremia and restricts HV K. pneumoniae to the liver, where they can be visualized as co-localized with KCs.

**Part II – Major Issues: Key Experiments Required for Acceptance**

Reviewer #1: Major Concerns

• In figures reporting the % of the bacteria per organ, the CFU need to be reported as well as percent. The overall differences in CFU are interesting and important in addition to the percent per organ. Specific questions I have: 1) do the hybrid K3 and K23 strains have similar or different bacterial burdens compared to K1 and K2? 2) Did splenectomy impact overall burden numbers?

• There is no mention in the Materials and Methods sections about replicate experiments. I noted that for many animal infections, the data are from a sample number of only 3. If this is indeed the complete data set, then the conclusions need to be softened. If these are representative experiments, then the M&M section must include information about reproducibility of the data.

• For the data generated with mice treated to reduce cell types, FACS analysis should be provided to indicate that the targeted cell types were indeed depleted. This is a necessary control to verify that the observed phenotypes are indeed due to depletion and not some other cause.

• A better definition of “capture” is desirable. Do you think KCs are rapidly phagocytosing all this bacteria or is it rapid binding followed by phagocytosis? Are there tissue culture assays that might help get a mechanistic idea of what this “capture” is?

• In Fig. 5. the pretreatment with purified capsule blocks the ability of KCs to ‘capture’ Kp is interesting but raises several questions.

-what happens to ∆cps? What is the explanation for how this strain is captured?

- if purified CPS only blocks capture of cognate Kp, does this mean there are receptors for lots of different capsule types?

- Does the lack of capture of KV Kp suggest that the K1/K2 CPS does not interact with KCs? The interpretation of the data in Fig 6F would seem to suggest KCs are capable of recognizing K1/K2, at least after prior exposure (see comment below regarding this experiment).

• The interpretation of the imaging in Fig 6F and movie S4 that KCs are responsible for capturing HV Kp is not convincing. There are numerous green dots not associated with the red staining, hinting that something else is capturing these bacteria. Staining for other cell types, or perhaps the use of the Clec4f-DTR mice could provide more convincing evidence.

Reviewer #2: -While the study uses capsule-switched recombinant strains, a strength, the only reported verification of these strains was through sequencing the recombined cps sequences. The study would benefit from additional verification of the production of capsule from these engineered heterologous strains, especially since they are largely associated with virulence-defective phenotypes.

Reviewer #3: 1) The authors do not validate that ATCC 43816 K3 and K23 strains produce capsule. If the isogenic mutants do not produce comparable levels of capsule to K2 or K1, that could explain the observed phenotypes.

2) Most animal studies are done with 3 mice, some only have 2, some have more. The small sample size must be justified for providing adequate statistical power. This could be achieved using power calculations.

3) Statistical analyses are poorly described and often not reported in the figure legends. It is not reported what statistical result is represented by the ****. It is hard to distinguish which results are non-significant vs not tested. And a lot of conclusions are made based on changes not reported to be statistically significant.

All data are plotted as mean ± SEM according to the methods, but that is not reported in the figure legends. SEM does not provide information about the scatter of the data, only where the true mean lies and this seems a bit misleading considering most data points have only been performed in triplicate, once.

4) There are major issues with the transparency of how the data was calculated, challenging the interpretation of the data. For example, it is unclear how the denominator is calculated when % total bacteria is presented. How are total bacterial CFUs in a mouse determined when considering both solid organs and blood? At minimum, absolute CFU counts should be included in the supplement. However, it may help with clarity if instead absolute CFU counts for each organ/blood be presented in the figures and the changes be reported as fold change. Evaluating percentages for CFU data is challenges as a 99% decrease is only a 2-log decrease, so a lot of resolution is lost when using percentages.

Some specific challenges related to this data presentation as it stands: 1) It is hard to tell if the total liver counts are the same across bacterial species and altered blood counts are changing the liver percentages or are there actually less bacteria in the livers during HV infection. 2) Absolute counts would also be important in the splenectomy data as one of the organs contributing to the denominator is missing in half the groups. 3) % inoculum in 3H makes 1-log increase in bacteria look huge compared to a 1-log decrease.

Other issues with transparency in describing how the data were analyzed are below:

Time 0 CFU/ml in blood – is this the inoculum CFU/ml or the actual CFU/ml in the blood? It’s surprising that the time 0 CFU/ml in blood would be the same CFU/ml as the inoculum. I would expect it to be lower as the inoculum is diluted in the blood.

Fig 3I and 3K – the % inoculum in K doesn’t match the absolute counts in I. I would expect the ∆cps % inoculum to be 10^5 CFU/ml / 10^8 CFU/ml x100% = 0.1%

When discussing changes “by XX%” it hard to know how that number was calculated. It might be clearer if these are reported as fold-change relative to the control condition.

5) Lines 350-355 and throughout the results section: The conclusion that Kupffer cells capture LV capsule types based on the CPS structure, isn’t fully supported by the data presented. How can this statement be supported by the evidence that often the ∆cps strain phenocopies LV strains without having a CPS?

**Part III – Minor Issues: Editorial and Data Presentation Modifications**

Reviewer #1: Minor Concerns

• In general, I would recommend slightly larger figures, or at least larger text. Several were difficult to read due to the small text.

• Reference given for ATCC 43816 (line 122 Ref #36) does not seem correct?

• For some panels with infection data, the number of mice used is indicated, but for others this information is lacking. Please include for all. Figs. 3I-K indicates only 2 mice used? Cannot accurately do statistics with only 2 mice, so please remove asterisks if this number is correct.

• It is curious that the HV strains had a larger proportion of bacteria in the spleen than in the liver. As K1 strains are known to cause liver abscesses, some speculation on this result in the Discussion would be worth noting.

• Lines 194-195, an important clarification here is that LV bacteria have an advantage over acapsular bacteria.

• Line 215, please expand the description of the Clec4f-DTR mice for readers not intimately familiar with this mutation/DTR treatment.

• Lines 230-231, the statement that the ability of KCs to capture acapsular and LV Kp but not HV Kp explains the difference in these pathotypes is overstated. This is unlikely to be the only reason for the differences. Please make the language more suggestive than definitive.

• It is assumed from Fig. 3 and beyond, that mostly the hybrid strains are used. It might be helpful to give these strains names to indicate that. The data in Fig. 5 uses LV isolates, but there is no change in nomenclature in the Results to distinguish this.

• Line 366, referring to comment above about Fig 6F, this language should be altered. Also, “glued” is not the best term for bacteria interacting with host cells.

• Line 415, please state the route of infection for this LD50

• In the Mouse infection section of the Materials & Methods, please provide some brief detail about the inoculations and surgeries. Please do not require your readers to find other papers to find the basic info for these procedures!

• In the paragraph about preparation of inactivated bacteria for immunization, it would be helpful to include culture volume and final resuspension volumes, as well as inoculation volume for the killed bacteria.

Grammar & typos

Throughout, in describing fold change, “fold” is not plural

Line 142, perhaps “recovered” in place of “regrew”

Line 202, We, not W

Line 342, I’m not sure use of the word “act” is appropriate for the role of capsules. They provide protection, but not “act” on circulating phagocytes.

Line 363, inactivated

Line 378, “trial and error”

Line 482, ATCC 43816

Reviewer #2: -Regarding the conclusions with the data using Clec4f-DTR mice, the authors state that the difference in hepatic capture "explains the drastic difference in virulence traits between these two types of the pathogen."

In addition to causing enhanced bacteremia levels and less early hepatic capture, does selective KC depletion with diphtheria toxin in transgenic mice allow LV strains to show the expected enhanced virulence (eg lower survival rate of infected mice) after the infection progresses past the early 30 minute window studied?

-If recombinant K1 or K2 HV capsule can be switched into a LV strain, can this mediate reversal of KC clearance and HV?

-The possibility that the complement system, and different degrees of opsonization by complement with different capsule types, is a driver of different liver capture phenotypes should be discussed (in addition to the other mechanisms proposed by the authors in lines 350-357). Complement binding to diverse K. pneumoniae capsule types has been documented (eg see Jensen et al 2020), and it is possible that certain capsule structures interact with complement to create efficient KC opsonins, while interaction with other capsule structures does not efficiently opsonize.

Regarding the potential for complement interactions as a determinant of capture of invading bacteria by the liver and spleen, the authors should consider citing Spiegelberg et al 1963 and Brown & Frank 1981 (Brown & Frank 1981 was cited in the manuscript but for a different purpose). These two papers also examined the role of liver vs spleen in rapid vs slow clearance of invasive organisms, another context in which these papers can be cited.

-Figs 3-4: While showing the different tissue burdens as percentages (eg. Fig 3 panels D, E, G) is clear and useful, the interpretation of these data could benefit by providing total absolute counts of bacteria with each strain in the liver. This was done in Fig. S2A for the 5 min time point and was very useful. A similar supplemental graph with absolute liver counts at 30 minutes would be helpful to aid in understanding if there are still high absolute bacterial counts in the liver (as would be inferred since there are such high bacterial loads in bloodstream with HV strains at 30 min and beyond -- Fig. 3B,C).

-The implication of this manuscript's findings for the propensity of certain HV K. pneumoniae infections to lead to pyogenic liver abscess could be discussed.

-Line 311 states "lethality of mice when high infection doses of LV Kp are used...", but mouse lethality/survival data were not shown in that context. This should be provided or the text revised.

-Line 816: Fig. 3H legend: "host killing" could lead to confusion since the graph title states "bacterial killing"; the text should be revised (eg, bacterial killing; immune killing)

-line 269: should "3.1 days" be 3.5 days, per the figure?

-S2C and D Fig (lines 32-26): The legend should be improved to clarify what is being shown. Legend to S2C states "Percentage of bacterial clearance in blood circulation of mice post infection as in A", but A does not show percentages, it shows absolute bacterial counts. The legend to panel D is confusing/non-intuitive perhaps because it describes counting killed bacteria (rather than counting viable bacteria), so the wording could be made clearer to describe what was actually measured and shown.

References:

Jensen TS et al, Microbes Infect. 2020 Jan-Feb;22(1):19-30. doi: 10.1016/j.micinf.2019.08.003

Spiegelberg HL et al, J Immunol. 1963 May;90:751-9.

Brown EJ et al. The role of complement in the localization of pneumococci in the splanchnic reticuloendothelial system during experimental bacteremia. J Immunol. 1981; 126(6): 2230-5.

Reviewer #3: Line 92 states that how capsule affects pathogenesis is uncharacterized, but the prior paragraph describes several mechanisms by which capsule affects pathogenesis.

Line 97: A little more background information about Kupffer cells in the introduction would help readers understand the rationale and impact of the subsequent findings.

Line 111: Please more precisely define 50% mortality. I think the value used is the time at which half the mice died, but it isn’t clearly stated.

Fig 1C and E; Fig 2C. Is there attrition of sample numbers at 12+ hr? Or does each dot truly represent 3 mice? If there is attrition of sample numbers, how was that accounted for in the statistical analyses?

Line 164 states that the spleen was the organ with the highest burden of HV Kp, but it appears that sometimes it is the liver is in Fig. 3E.

Fig 3H would benefit from plotting the X-axis in scale, as in 3C and 3I.

Fig 4E and G: the Bacteria per FOV – are these absolute values or percentages? Is the bar graph in 4F supposed to be titled “bacteria on KC”? There is a discrepancy between the figure legend and line 228 of the text.

Fig 6 – Are naïve mice immunized with vehicle control or no treatment?

The methods state that CD1 or C57BL/6 mice were used in the experiments, but the figure legends do not specify which mice are used in which experiments.

PLOS authors have the option to publish the peer review history of their article (what does this mean?). If published, this will include your full peer review and any attached files.

Reviewer #1: No

Reviewer #2: No

Reviewer #3: No
---

## [Editor Report · Decision Letter 1]

22 Jun 2022

Dear Jing Ren,

We are pleased to inform you that your manuscript 'Capsule Type Defines the Capability of Klebsiella pneumoniae in Evading Kupffer Cell Capture in the Liver' has been provisionally accepted for publication in PLOS Pathogens.

Best regards,

Joan Mecsas

Associate Editor

PLOS Pathogens

Christoph Tang

Section Editor

PLOS Pathogens

Kasturi Haldar

Editor-in-Chief

PLOS Pathogens

orcid.org/0000-0001-5065-158X

Michael Malim

Editor-in-Chief

PLOS Pathogens

orcid.org/0000-0002-7699-2064
---

## [Editor Report · Acceptance letter]

3 Jul 2022

Dear Dr. Zhang,

We are delighted to inform you that your manuscript, "Capsule Type Defines the Capability of *Klebsiella pneumoniae* in Evading Kupffer Cell Capture in the Liver," has been formally accepted for publication in PLOS Pathogens.

Best regards,

Kasturi Haldar

Editor-in-Chief

PLOS Pathogens

orcid.org/0000-0001-5065-158X

Michael Malim

Editor-in-Chief

PLOS Pathogens

orcid.org/0000-0002-7699-2064